

# Design and Rocket Deployment of a Trackable Pseudo-Lagrangian Drifter based Meteorological Probe into the Lawrence/Linwood EF4 Tornado and Mesocyclone on 28 May 2019

Reed Timmer[1,*], Mark Simpson[1,*], Sean Schofer[1], and Curtis Brooks[1]

[*]These authors contributed equally to this work.
[1]Team Dominator, Golden, Colorado, USA

**Correspondence:** Reed Timmer (reedtimmer@gmail.com)

**Abstract.** A custom lightweight, miniaturized, and trackable meteorological probe was launched by a model rocket into the inflow region near an EF4, long-tracked tornado south of Lawrence, Kansas, on 28 May 2019 and sampled tornado core flow. The rocket reached apogee at 439 m AGL, releasing the "pseudo-Lagrangian drifter" by parachute directly into the tornado vortex. The probe reached a velocity of 85.1 m s$^{-1}$ in the first revolution around the tornado, measured a pressure deficit of -113.5 hPa

at 475 m MSL, and sampled a tornadic updraft of 65.0 m s$^{-1}$. The probe then transitioned to an environment exhibiting more tilted ascent above an altitude of 4,300 m MSL at speeds up to 84.0 m s$^{-1}$ to a maximum altitude of 11,914 m MSL. 1 Hz pressure, temperature, relative humidity, GPS, acceleration, gyroscope, and magnetometer data for the flight were transmitted in real-time to a ground station until the probe landed 51 km northeast of the launch position. The probe was recovered without damage, which is attributed to the pseudo-Lagrangian Drifter design, and then higher resolution 10 Hz data was downloaded.

This novel deployment method and design facilitate data collection in real-time from within tornadoes, the mesocyclone, and downdraft without requiring the probes to be recovered or for researchers to enter the circulation to deploy equipment.

## 1 Introduction

Supercell tornadoes are the most damaging local severe weather phenomena capable of producing wind speeds greater than 100 m s$^{-1}$ and atmospheric pressure deficits greater than 100 hPa (Blair et al., 2008; Karstens et al., 2010). The strongest,

most damaging winds of a tornado typically occur within the core flow or radius of maximum winds (RMW) along the outer boundary of the tornado core of low atmospheric pressure (Karstens et al., 2010; Wurman et al., 2013). This ring of maximum winds of a tornado is driven by strong horizontal and vertical pressure gradients at the outer periphery of the inner core near and just above the surface (Snow et al., 1980; Lewellen et al., 2000; Fiedler, 2009; Wurman et al., 2013). Direct measurements of above-ground winds and thermodynamics from within the core flow of a tornado had not been accomplished until the present

study due to the extreme atmospheric conditions prevalent within the RMW (Wurman et al., 2013).

The authors and others (Snow et al., 1980; Lewellen et al., 1997, 2000; Fiedler, 2009; Karstens et al., 2010; Wurman et al., 2013) theorize that the most significant pressure deficits in tornadoes occur in the core within the RMW very near and just above the ground, often exciting strong, jet-like updrafts that are capable of causing damage. High-resolution cameras and



drones have documented these damaging updrafts of tornadoes in recent years. The present study's deployment of miniaturized,
pseudo-Lagrangian sensors into tornadoes by rocket can measure the horizontal and vertical winds and pressure perturbations
within the tornado's RMW and core.

Direct measurements of atmospheric conditions from inside tornadoes have been limited to ground-based instrumentation (Samaras and Lee, 2004; Blair et al., 2008; Karstens et al., 2010; Kosiba and Wurman, 2013; Wurman et al., 2013). The first ground-based, deployable instrument pack or probe for in situ measurements of meteorological data inside tornadoes was constructed in 1980 and called the TOtable Tornado Observatory (TOTO) (Bedard Jr and Ramzy, 1983; Bluestein, 1983, 1999). The usage of TOTO was abandoned when it was determined that placing it directly into the path of a tornado was challenging (Bluestein, 1999).

Tim Samaras recorded a pressure deficit of 100 hPa inside an F4 tornado in Manchester, South Dakota, on 24 June 2003 using the Hardened In Situ Tornado Pressure Recorder (HITPR) (Samaras and Lee, 2004; Karstens et al., 2010). In Tulia, Texas, on 21 April 2007, a mobile mesonet vehicle was struck by an EF2 tornado and recorded a pressure deficit of 194 hPa along with a maximum wind velocity of 50.4 m s$^{-1}$ (Blair et al., 2008). Ground-based measurements of wind and pressure deficit inside a tornado are essential because the surface conditions are responsible for structural damage and human impact. However, ground-based measurements alone do not provide information on the three-dimensional wind, thermodynamics, or structure of tornadoes.

Stirling Colgate of Los Alamos National Laboratory was the first to attempt measurements of meteorological data above-ground and inside tornadoes with instrumented lightweight (<1 lb) rockets launched from a small aircraft (Colgate, 1982). The rocket was fired horizontally at a velocity close to Mach 1 to transect the tornado at altitude while measuring pressure, temperature, ionization, and electric field at 1-meter intervals. During over 120 hours of flying, only one tornado was observed, and the four rocket launches missed the target. Additionally, it was determined that the equipment needed to be hardened and engineering improved. This contrasts with the present study, which uses a rocket launched from the ground as a delivery mechanism for a parachuted, pseudo-Lagrangian sensor.

Over the past two decades, field experiments such as VORTEX, VORTEX2, and Project TORUS have improved the understanding of supercells, tornadoes, and particularly tornado environments using mobile Doppler radar (X-, W-, Ka-band), in situ ground-based probes, balloons, mobile mesonets, and Unmanned Aerial Systems (UAS) (Straka et al., 1996; Bluestein et al., 2003; Wakimoto et al., 2003; Bluestein et al., 2004; Samaras and Lee, 2004; Blair et al., 2008; Weiss and Schroeder, 2008; Karstens et al., 2010; Kosiba and Wurman, 2010, 2013; Wakimoto et al., 2011; Wurman et al., 2012; Tanamachi et al., 2013; Winn et al., 1999; Samaras, 2004; Pazmany et al., 2013; Frew et al., 2020; Houston et al., 2020; Markowski et al., 2018).

Mobile radars are mainly limited to measuring horizontal wind of storms and tornadoes, with the vertical component being inferred due to inclined measurements unless the measurements are taken vertically from dangerous positions inside a tornado. The UAS technology of Project TORUS is not intended for direct measurements inside of a tornado core flow (Frew et al., 2020; Houston et al., 2020).

Multiple-elevation mobile radar data coupled with photogrammetry techniques and or ground-based wind measurements have successfully derived information on the three-dimensional winds of tornadoes (Wakimoto et al., 2011; Kosiba and Wur-





man, 2010, 2013; Tanamachi et al., 2013). However, mobile radar-based analyses of winds inside tornadoes using multiple
elevations are limited in spatial and temporal resolution, range and provide limited information on the thermodynamics of
tornadoes (Markowski et al., 2018).

Using pseudo-Lagrangian drifters for infiltrating thunderstorms and supercells with sensors has been accomplished using
a dual-balloon approach (Markowski et al., 2018). The thermodynamics of supercell storms (and tornadoes) are essential
to understand because wind accelerations and storm morphology are driven primarily by pressure gradient and buoyancy
forces (Markowski et al., 2018; Bartos et al., 2022). The wind speeds and accelerations inside a tornado should also be driven by
pressure gradient and buoyancy forces. A tornado has yet to be directly sampled by the dual-balloon deployments of Markowski
et al., 2018; Bartos et al., 2022.

The present study's authors have tested many airborne delivery mechanisms for deploying meteorological sensors into
tornado core flows over the past two decades. These include air cannons, parachuted sensor packs dropped by remote-controlled
aircraft, and ground-deployed inflatable plastic bubbles to be ingested by the surface inflow of a tornado. After experimenting
with many different delivery methods, it was theorized that to obtain meteorological data by a sensor from within a tornado core
flow, the speed of model rockets would be required to penetrate subsiding air that is often proximal to a mature tornado (Lemon
and Doswell III, 1979). Descending air is often nearby or even surrounding a supercell tornado because these tornadoes most
often occur near strong gradients of vertical velocity (Lemon and Doswell III, 1979).

This study presents the design and successful deployment by a rocket of a pseudo-Lagrangian drifter into a tornado vortex
and mesocyclone with an overview analysis and discussion of the data received. This novel methodology is presented as a
solution to measure directly the three-dimensional wind and thermodynamics inside a tornado.

## 2   Methodology

### 2.1   Probe Design

The authors designed and built a custom-engineered probe (Fig. 1) and firmware for the present study. Real-time data reception
and location tracking were deemed necessary to ensure that the loss of the probe due to the hostile environment or inability to
recover did not impact data collection. The software was designed to permit multiple probes to be airborne concurrently.

This lightweight probe has onboard miniaturized sensors for pressure, temperature, humidity, GPS, and a 3-axis Inertial
Measurement Unit (IMU). The rocket's pseudo-Lagrangian flight characteristic and small payload capacity necessitated a
compact surface-mount design (2.5 x 7.6 cm) with a total mass of 30 g (17 g plus 13 g battery). The probe's onboard sensors
were carefully selected based on criteria for wide environmental operating ranges, sampling rates, accuracy, mass, and power
efficiency (Table 1). A high sampling rate of 10 Hz was compatible with all sensors, excluding the GPS, which has a maximum
rate of 5 Hz.

The GPS sensor selected for the probe was the ORG1411 "Nano Hornet" by OriginGPS. This device has a small form factor
and mass while maintaining a GPS fix in accelerations of up to 39.2 m s$^{-2}$. This sensor has an inbuilt patch antenna which
provides weight savings compared to an external antenna.





**Figure 1.** 3D render of a custom-engineered probe showing the sensors and major electronic components used in the design.

Similarly, the BME280 sensor by Bosch Sensortec was determined to be an ideal barometric pressure and humidity sensor, given its small mass, size, and performance. The higher pressure sampling rate was advantageous for detecting gravity waves or other fine-scale pressure oscillations within the tornado and parent mesocyclone by comparing GPS altitude to pressure.

However, it was determined that the temperature measurement could be improved with the addition of the SI7053 temperature sensor by Silicon Labs. This provided improvements in accuracy and thermodynamic response time ($\tau_{63\%}$) over the BME280 (Table 1).

An IMU was included on the probe to measure the forces and accelerations experienced inside the tornado and the probe's orientation. The 9-axis BNO085 is a miniature IMU by Hillcrest Labs that provides this data from an incorporated 3-axis

magnetometer, accelerometer, and gyroscope. The miniature size and mass of the BNO085 are ideal for flight within a tornado. Using quaternions from the IMU and the sensor-fusion, the Tait-Bryan angles for pitch, yaw, and roll (as is used in aircraft orientation) were derived. Information about orientation and forces imparted on the probe help identify turbulence and transitions between shear layers. After manufacturing, a software routine was developed to calibrate the IMU and store the data on the



**Table 1.** Probe sensor summary.

| Measurand | Sensor Model | Sampling Rate (Hz) | Mass (g) | Dimensions (cm) | Operating Range (°C) | Specifications |
|---|---|---|---|---|---|---|
| GPS | ORG1411 | 5 | 1.4 | 10.0×10.0×3.8 | -40 to +85 | velocity: <600 m s$^{-1}$ |
| | | | | | | acceleration: ±39.2 m s$^{-2}$ |
| | | | | | | positional accuracy: |
| | | | | | | horizontal: <2.0 m |
| | | | | | | vertical: <3.5 m |
| Pressure, RH | BME280 | 10 | 0.009 | 2.5×2.5×0.9 | -40 to +85 | pressure: $\geq$ 300 hPa |
| | | | | | | RH response($\tau_{63\%}$): 1 s |
| | | | | | | accuracy: |
| | | | | | | pressure: ±1.7 hPa |
| | | | | | | RH: ±3% |
| Temperature | SI7053 | 10 | 0.003 | 3.0×3.0×0.7 | -40 to +125 | response($\tau_{63\%}$): 0.7 s |
| | | | | | | accuracy: ±0.3 °C |
| IMU | BNO085 | 10 | 0.043 | 5.2×3.8×1.1 | -40 to +85 | linear acceleration: ±79.5 m s$^{-2}$ |
| | | | | | | gyroscope: ±34.9 rad s$^{-1}$ |
| | | | | | | magnetometer |
| | | | | | | ±1,300 $\mu$T (x,y) |
| | | | | | | ±2,500 $\mu$T (z) |
| | | | | | | accuracy: |
| | | | | | | lin. acceleration: ±0.35 m s$^{-2}$ |
| | | | | | | gyroscope: ±3.1 ° s$^{-1}$ |
| | | | | | | magnetometer: ±1.4 $\mu$T |

electrically erasable programmable read-only memory (EEPROM), as stresses built up in the soldering process can offset the
factory calibration of micro-electromechanical systems (MEMS) sensors on the die.

The probe used Long Range (LoRa) radio transmission to transmit and receive data to and from a ground station. LoRa
is a spread spectrum technology that can still work even when the signal is below the noise floor. The power required is
also minimal compared to other technologies, and throughput is traded against range, with longer distances requiring less
throughput (Augustin et al., 2016; Noreen et al., 2017). Transmission occurs in the Industrial, Scientific, and Medical band
(ISM) at 915 MHz and does not require licensing. Point-to-point LoRa was chosen over the more commonly used wide area
star-of-stars topology Long Range Wide Area Network (LoRaWAN) (Semtech, 2019) due to the increased throughput and





ability to maintain transmission in sparsely populated areas. A star-of-stars topology is one in which gateways relay messages between devices and a centralized network server.

Time-Division Multiple Access (TDMA) was implemented with regular framing pulses from a ground station, allowing multiple probes to be tracked simultaneously. 1 Hz data transmissions allow the complete sensor data to be transmitted concurrently over LoRa for nine airborne sensors for distances of up to 100 km. The TDMA protocol was modified such that non-reception of the framing pulse did not result in an immediate loss of synchronization and that opportunistic reception could continue in the event of a loss of synchronization.

A 1/4 wave monopole antenna was chosen for the probe and coiled to reduce size. This antenna configuration allowed for an omnidirectional radiation pattern such that the probe's orientation in flight would not impact the communication range.

The electronics were waterproofed using silicone conformal coating and a polytetrafluoroethylene (PTFE) barrier over the port on the BME280 to avoid damage to the electronics and unpredictable behavior due to current leakage. Power was provided by a small rechargeable 350 mA·h Lithium-Ion Polymer (LiPo) battery.

## 2.2 Ground Station Design

The mobile ground station consisted of a Raspberry Pi Zero W, and a custom LoRa daughterboard with an inter-integrated circuit ($I^2C$) organic light-emitting diode (OLED) display and antenna. A 1/4 wave monopole omnidirectional antenna was used instead of an increased range directional antenna to simplify tracking the probe in a severe weather environment.

The station contained custom software on an SD Card and provided a web server over a Wi-Fi network. In addition, the station broadcasted the TDMA framing pulse, handled the communication, stored data from the sensors, and provided the user interface for monitoring the sensors in real-time.

Accessing the interface was via a web browser on a mobile device or computer, and an Apple iPad was used for this purpose. The bearing and distance were calculated from the chase vehicle to the probe by comparing the onboard GPS location of the iPad to the GPS location received from the probe. A 10 A·h rechargeable battery pack provided power.

## 2.3 Rocket and Launcher Design

A custom-modified Magnum Sport Loader dual motor model rocket from Quest Aerospace was used to launch the probes into the tornado (Fig. 2). The dual-motor design enhanced thrust, range, and payload capacity.

High velocity was required to pass through the subsidence often situated nearby an occluded tornado (Lemon and Doswell III, 1979); therefore, higher thrust class D Ammonium Perchlorate motors (Aerotech D21-7T 18mm) were used for each of the dual motors. The motor uses Blue Thunder propellant, has a mass of 25 g, and a peak thrust of 32.1 N. Due to a long 7 s delay between engine burnout and the parachute ejection charge, the rocket coasts further and can cover the distance required to transit into the tornado. The upper three stabilization fins were raised higher on the rocket body to improve stability in high winds. A single hand-sewn nylon parachute with an outside diameter of 0.61 m was used to provide resistance to debris and damage from the ejection charge while lofting in vertical high-velocity winds.



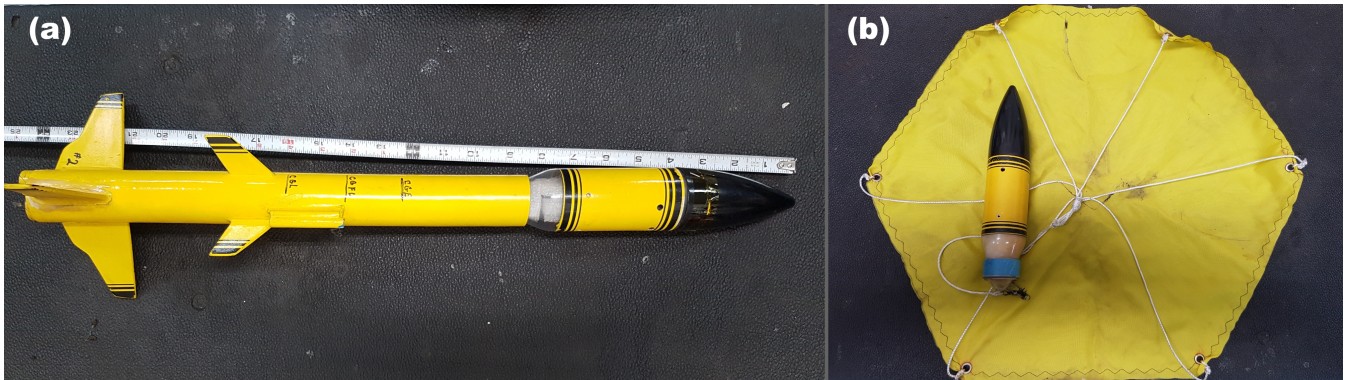

**Figure 2.** Customized rocket (a) and pseudo-Lagrangian drifter deployment configuration (b).

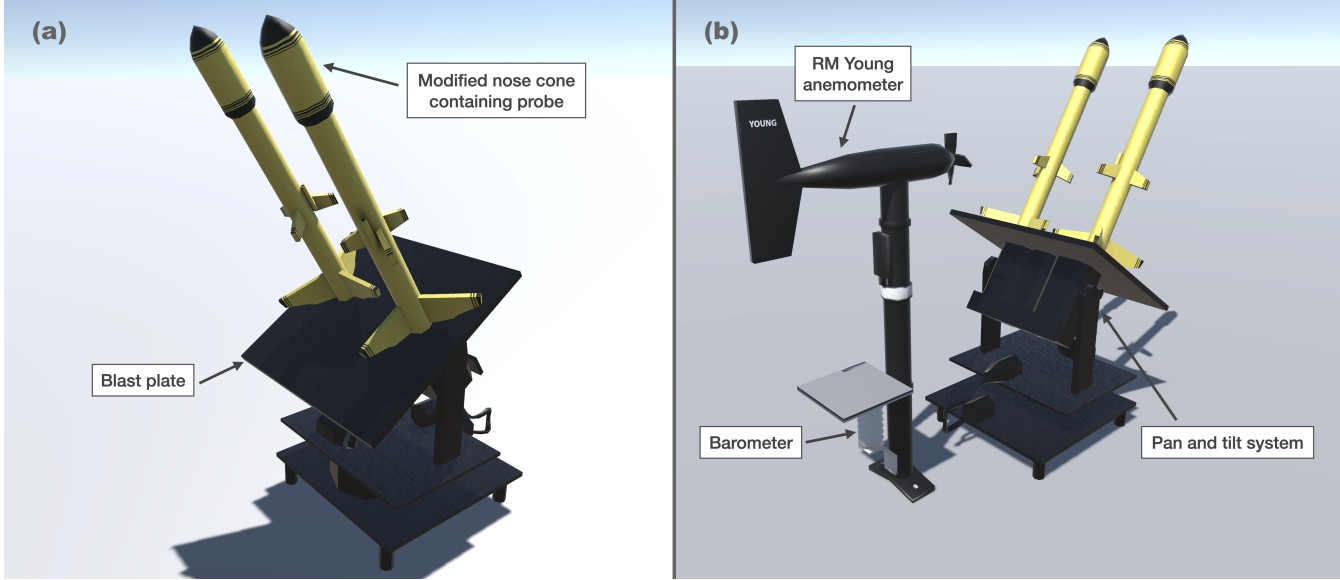

**Figure 3.** 3D render of rooftop rocket launcher (credit: OUTBRK). Rockets and blast plate (a); weather station and rocket launcher pan/tilt mechanism (b).

Rocket flight simulator software (RockSim Version 9) was used to optimize the center of pressure and achieve maximum flight range without tumbling. Simulations also tested how the rocket would respond to substantial winds and atmospheric conditions found in the inflow zone of a tornado. The maximum payload capacity of this customized design was determined to be 50 g with a parachute descent rate of approximately 4 - 6 m s$^{-1}$, depending on air density. The probe was housed within the nose cone compartment with expanded polystyrene foam for thermal protection.



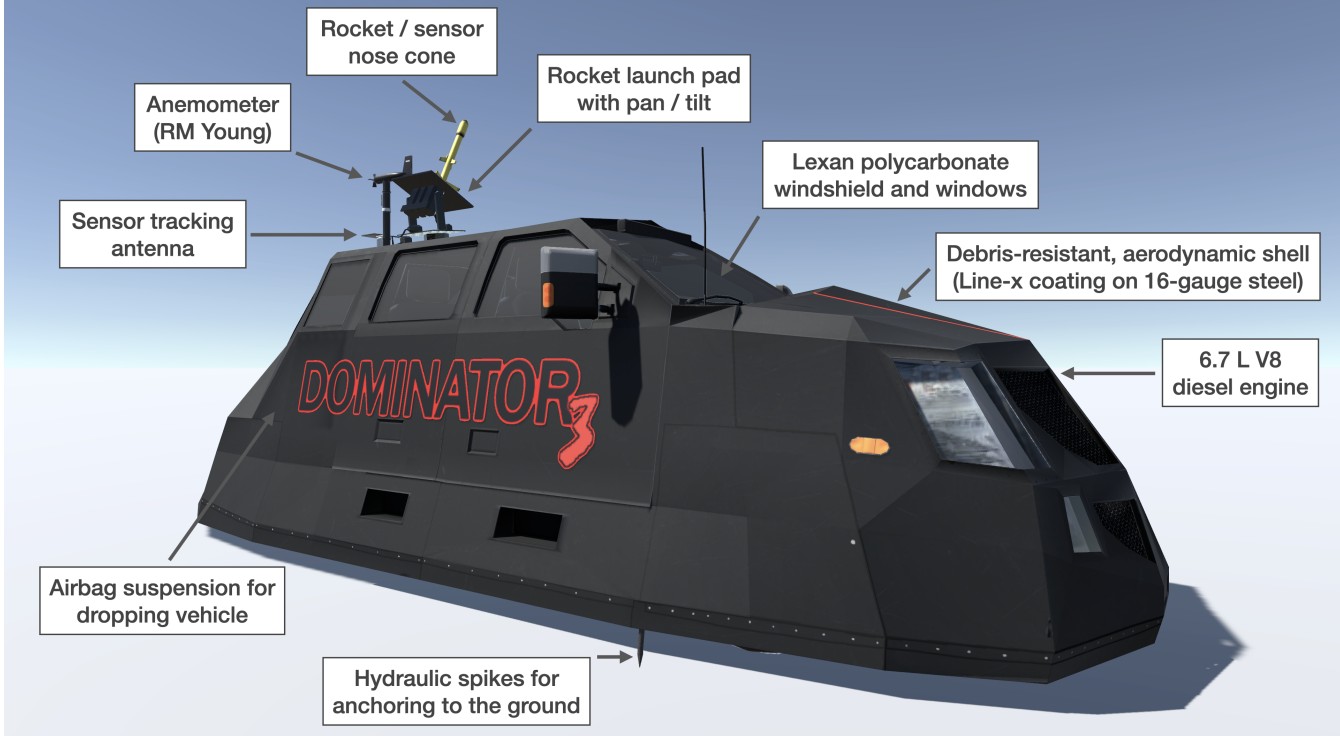

**Figure 4.** Dominator 3 storm chasing vehicle (credit: OUTBRK).

A mobile rocket launcher (Fig. 3) with pan-and-tilt capability was engineered and mounted to the roof of the armored chase

vehicle Dominator 3 (Fig. 4). The motor and gears of the launcher were derived from the automated window lift system of a junkyard vehicle. The design incorporated a substantial blast plate and two 0.91 m launch rods to launch multiple rockets.

The potential exists for a return lightning strike from a rocket's ionization trail, so a custom control panel was designed inside the Dominator 3 to aim and launch the rockets from within the protective armor (Fig. 5).

## 2.4 Launch Safety

The low mass of the rocket, propellent, and payload combined with the engine size selection results in a Federal Aviation Administration (FAA) Class 1 amateur model rocket designation (Federal Aviation Administration, 2022). Regulations and recommendations for Class 1 rockets in the USA are governed by the FAA, National Fire Protection Association (NFPA), and National Association of Rocketry (NAR).

(Federal Aviation Administration, 2022) specifies that a Class 1 amateur rocket must not be launched within a prohibited

or restricted area unless permission is obtained, which is determined with aviation maps. Additionally, the rocket must be suborbital, not cross into a foreign nation, be unmanned and "not create a hazard to persons, property, or other aircraft."





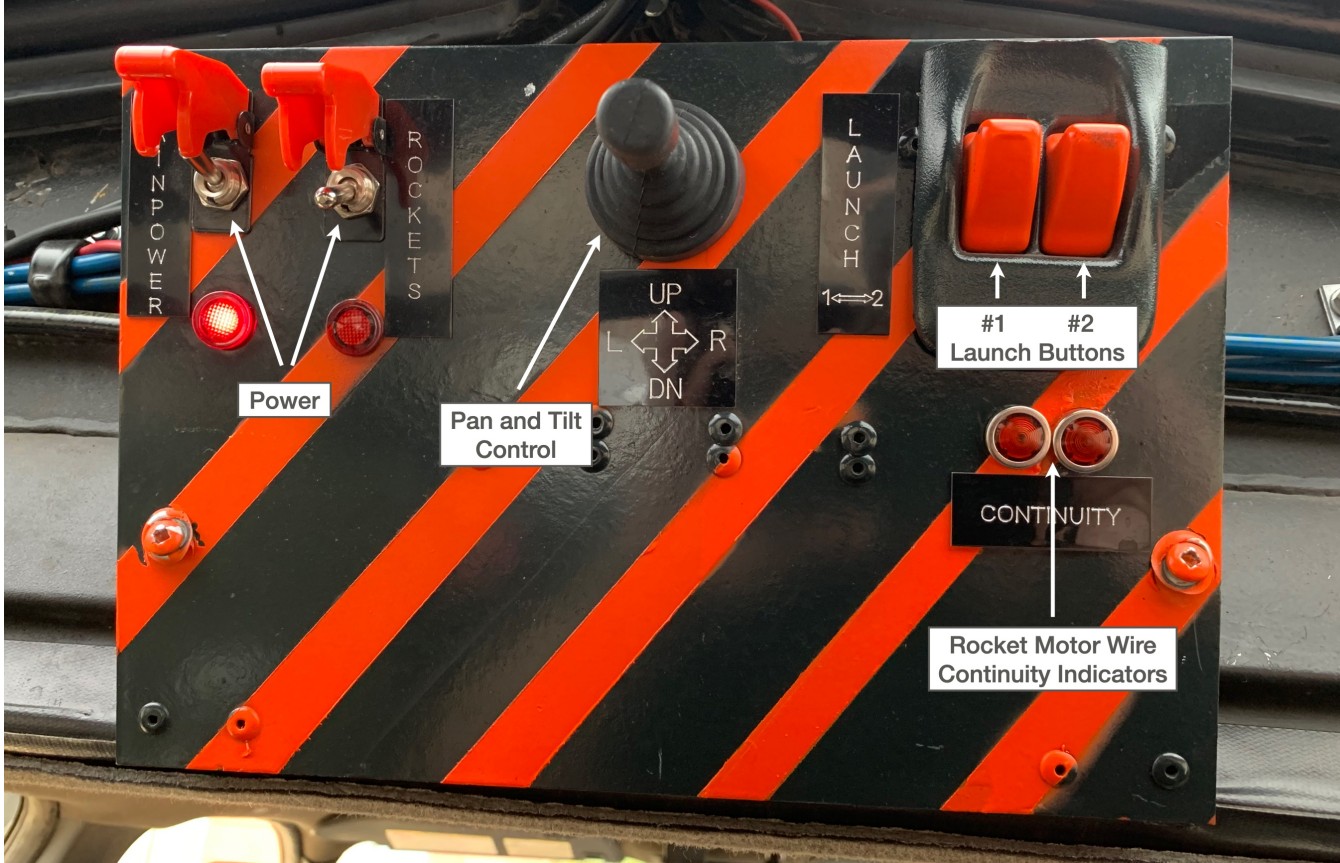

**Figure 5.** Custom control panel for launching rockets.

The NFPA recommendations are adopted at a state level by most states. The minimum site dimension(s) for a Class D motor are 152 m and should be "in a cleared area, free of tall trees, power lines, buildings, and dry brush and grass". A launch of this small size requires a spectator distance of 9 m (National Fire Protection Association, 2018).

NAR summarizes the NFPA recommendations and specifies that the site "must be free of tall trees, power lines, buildings, and dry brush and grass. The launcher can be anywhere on this site, and the site can include roads. Site dimensions are not tied to the expected altitude of the rockets' flights" (National Association of Rocketry, 2022b, a).

    The risk to aviation is negligible as pilots avoid tornadoes and thunderstorms. The low payload mass of 30 g is less than a North American Robin, so the impact would be similar to a small bird strike in the improbable event of a collision. After

deployment, the probe and parachute behave similarly to other lightweight debris lofted in a tornado and which are at altitude. The descent phase is controlled by a parachute and does not present an impact hazard.

    FAA, NFPA, and NAR have accounted for the risk to people, property, and aviation according to the class definitions they have applied to the hobby and have defined regulations appropriate for safety. Many amateur rocket launches of considerably





larger sizes occur daily in the USA, and model rocketry has an excellent safety record. Hitting a vehicle or person is unlikely,
and damage at this size and mass is expected to be negligible.

In addition to fulfilling the legal requirements for rocket launches, we also model, sufficiently test rockets, and inspect
rockets before launch. The authors keep fire extinguishers available, team members include a firefighter and a fire marshal, and
launch site safety is assessed.

## 3 Deployment

### 3.1 Synopsis

The period between 17 May to 30 May 2019 was a prolific period for severe weather across the United States, with 456 torna-
does (NOAA National Centers for Environmental Information - NCEI, 2019b) and three tornado outbreaks (NOAA National
Centers for Environmental Information - NCEI, 2019a). Twenty-two were rated EF3, and two were rated EF4, including the
Lawrence/Linwood tornado (NOAA National Centers for Environmental Information - NCEI, 2019b).

On the morning of 28 May, a robust upper-level trough with 500 hPa temperatures below -20 (°C) covered a large portion
of the Mountain West region of North America. A strong southwesterly mid and upper-level jet stream extended downstream
over the southern Great Plains. The southerly low-level jet stream (LLJ) of greater than 20 m s$^{-1}$ (40 knots) maintained
deep low-level moisture up to a stationary front across northern Kansas and Missouri. An outflow boundary (OFB) from a
nocturnal mesoscale convective system (MCS) was draped from northwest to southeast across northeastern Kansas through
peak heating (Fig. 6) (Storm Prediction Center, 2019).

A dryline would advance east to central Kansas and Oklahoma as an additional focus for supercell development by late
afternoon. Beneath the strengthening LLJ, the backed surface winds associated with the OFB in a narrow zone between Kansas
City and Topeka would enhance 0-1 km storm-relative helicity to greater than 230 m$^2$ s$^{-2}$; this environment was conducive to
violent tornadoes during late afternoon and evening (Fig. 6).

The Storm Prediction Center of NOAA upgraded portions of northeastern Kansas and northern Missouri to a moderate
risk area for convective storms in the 13Z day 1 convective outlook with the potential for strong-to-violent tornadoes (Storm
Prediction Center, 2019). An enhanced risk area for convective storms was active to the west and southwest of the moderate
risk area for a dryline supercell mode but with lower tornado probabilities given weaker low-level wind shear. Twenty-five
tornadoes occurred in the United States on 28 May 2019, of which ten were in Kansas.

### 3.2 Tornado Event

The parent supercell of the Lawrence/Linwood tornado initiated in the open warm sector over the elevated terrain of the Flint
Hills at around 20:00 UTC and organized on approach to the OFB located just southwest of Kansas City. The first mesocyclone
occlusion yielded a brief EF2 tornado at 22:55 UTC, followed immediately by the rapid development of a more significant,
stronger tornado that began in southwestern Douglas County at 23:05 UTC.





**Figure 6.** Meteorological synopsis for the May 28 May 2019 tornado event.

The tornado quickly grew to over 1 km wide, producing EF3 damage within ten minutes of tornadogenesis along United States Highway 59 just south of Lawrence. The National Weather Service damage survey confirmed a 51.2 km long track through Douglas and Leavenworth Counties. Peak EF4 damage occurred in Linwood, with maximum winds estimated at 76 m s$^{-1}$ (National Weather Service, 2019) and a damage path width of over 1.6 km (Fig. 7). The tornado dissipated near the town of Bonner Springs, just west of the Kansas City International Airport.

The supercell was of the high-precipitation (HP) class. The authors had learned from previous experiments that the narrow zone of inflow into the northeast quadrant of a tornado would frequently be free of destructive hydrometeors that could damage the rocket (Fig. 8). The ideal launch location is also just ahead of the wind shift of the RFD gust front to ensure relatively



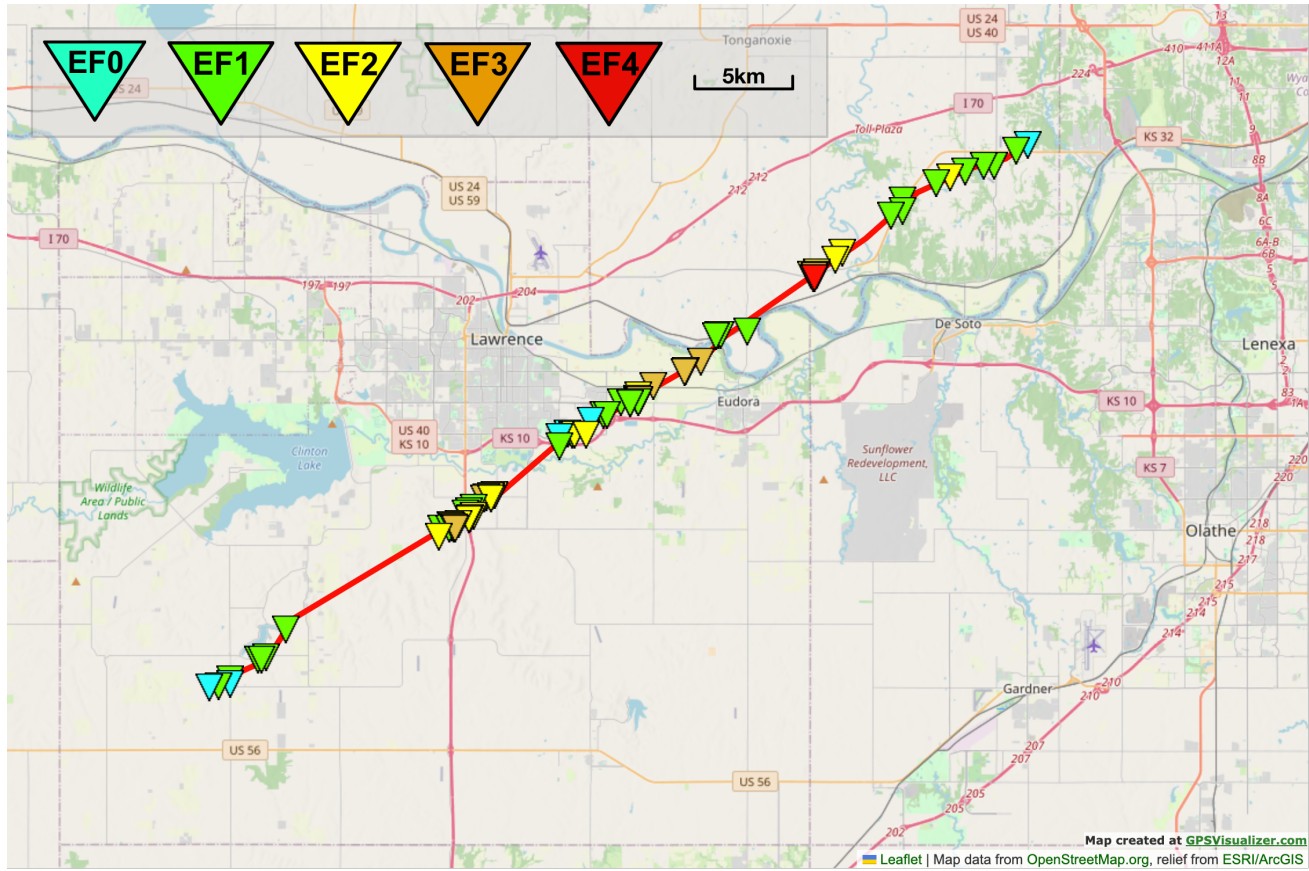

**Figure 7.** Damage and intensity track of Lawrence/Linwood tornado (credit: NWS).

calmer launch conditions (Lemon and Doswell III, 1979; Beck and Weiss, 2013). The easterly and northeasterly inflow winds at this location were more likely to transport the pseudo-Lagrangian drifter into a tornado.

Dominator 3 was positioned at the location of the marked asterisk (38.898911° N, 95.260437° W, and 258 m MSL) - (Fig. 8), and the rocket launcher aimed into the inflow region. At 23:17:33 UTC, the authors launched the rocket, and the payload was deployed. After launch, the tornado passed south of the launch location, with structural debris visible within the condensation funnel.

### 3.3   Rocket / Probe Flight

At 7.9 seconds after launch, 1 Hz real-time data revealed that the probe parachute deployed at 437 m AGL and entered the northwest side of the tornado. The crew then continued north on Highway 59 in Dominator 3 to maintain communication with the probe as the probe ascended in the tornado and mesocyclone. At 23:25:09 UTC, communication was lost as the probe approached the tropopause (horizontal distance from the launch of 10.4 km and 10,680 m MSL).





**Figure 8.** NWS Topeka radar (KTWX) 23:12 UTC; elevation 0.7°; county and state boundaries, rivers, and major roads are depicted; reflectivity and velocity in bottom-right (Lemon and Doswell III, 1979; Beck and Weiss, 2013) (credit: RadarScope/DTN).

The probe continued the flight and was recovered the next day on the grounds of a church in Leavenworth, KS (39.19° N, 225    94.94° W), having traveled 51.5 km northeast of the launch location. Subsequently, the authors downloaded the 10 Hz data from the onboard EEPROM for the entire flight.



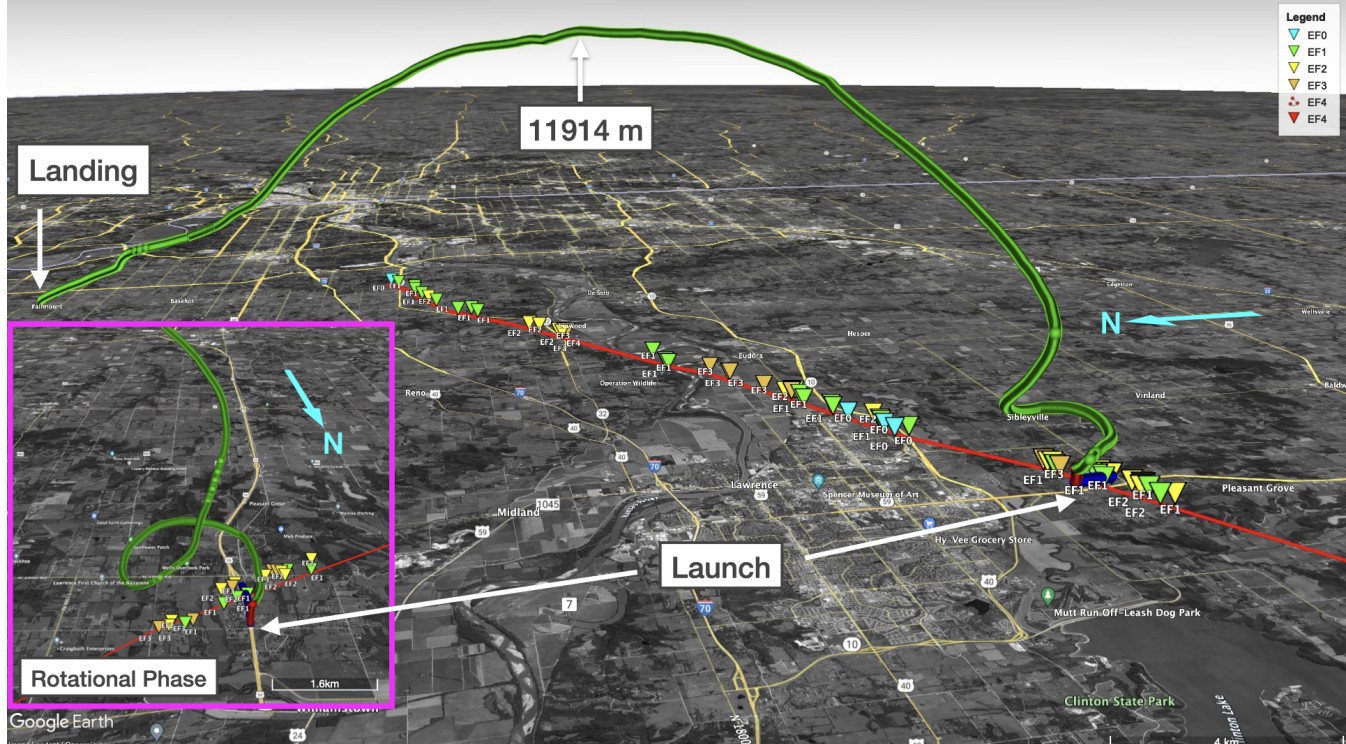

**Figure 9.** Probe GPS trajectory with NWS damage track superimposed and tornado close up (left panel). (© Google Earth).

## 4   Data, Analysis, and Discussion

### 4.1   GPS and Pressure

The probe recorded 5 Hz three-dimensional GPS data for time, horizontal position, altitude, speed, and heading. The GPS
velocity data recorded by the probe directly sampled wind speeds during the 30.2-minute flight inside the tornado and parent
mesocyclone, which is attributed to the probe being a pseudo-Lagrangian drifter following the parachute deployment phase.

The GPS trajectory of the probe's flight superimposed on the NWS damage track provides valuable data that could be used
for insight into the three-dimensional structure of the violent tornado and its parent mesocyclone (Fig. 9). It depicts the rotation
of the probe inside the tornado and then entrainment into the tilted updraft of the mesocyclone.

Owing to the availability of altitude data from the GPS and pressure data from the onboard sensor, it was possible to derive
a pressure perturbation metric to measure the pressure deficit between the ambient pressure and pressures inside the tornado
vortex and mesocyclone (Fig. 10e).

The GPS data recorded maxima for the three-dimensional speed of 85.1 m s$^{-1}$ (Fig. 10b), vertical velocity (w) of 65.0 m s$^{-1}$
(Fig. 10c), and a maximum altitude of 11,914 m MSL (Fig. 10a). The GPS Heading showed rotation within the tornado
(Fig. 10d).







**Figure 10.** GPS data and pressure perturbation sensor.

A storm motion of 21 m s$^{-1}$ from 220° was derived from radar and subtracted from the GPS position referenced to the launch location to produce an X/Y position relative to the storm. This was then used to plot storm relative GPS three-dimensional velocity and pressure perturbation (Fig. 11). The probe completed 1.5 revolutions around the tornado in near-circular movement with a diameter of 1.6 km (Fig. 11a), which is in agreement with the estimated maximum damage width of 1.6 km (National Weather Service, 2019). The 85.1 m s$^{-1}$ wind speed measured by the probe is in accordance with the 76 m s$^{-1}$ wind speed estimated by the damage survey (National Weather Service, 2019), and the pressure perturbation (Fig. 11b) indicates a deficit in the range of 80 to 113.5 hPa, which concurs with surface pressure deficits measured in other tornadoes (Samaras and Lee, 2004; Karstens et al., 2010; Blair et al., 2008). Additionally, the wind speed is greater in the southeast portion of the tornado and lower in the northwest, which is attributed to the northeasterly storm motion (Fig. 11a).



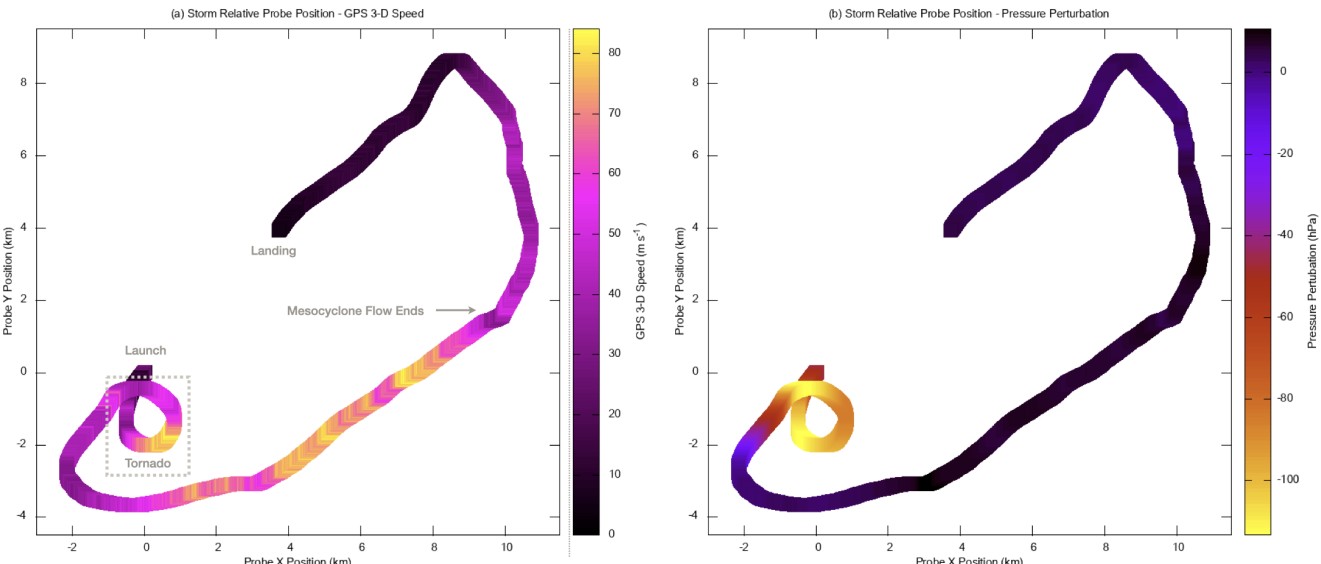

**Figure 11.** Storm relative probe position for the flight with GPS 3-D Speed (a) and Pressure Perturbation (b).

The 39.2 m s$^{-2}$ acceleration limit of the ORG1411 was surpassed for a short period during the powered rocket flight phase, which led to a loss of fix in two-dimensional position for 5 seconds and accuracy in altitude for 60 seconds. However, this still allowed for valuable data to be recorded in the near-ground layer of the tornado. The authors have since switched to a GPS receiver with a high dynamics mode that can reacquire a three-dimensional fix after rocket launch within 2 seconds for future deployments.

The operating range of the BME280 for barometric pressure extends down to 300 hPa per the Bosch datasheet; however, this flight reached a pressure altitude of 208 hPa. Subsequent cross-analysis between the GPS data and the Topeka 00Z meteorological balloon sounding confirmed the validity of barometric pressure readings below 300 hPa. Improvements to the design since this flight include an additional onboard pressure sensor with an operating range extending down to 100 hPa to serve as a second source of pressure data.

**4.2   LoRa**

During the probe flight, the ground station was handheld outside the side window of Dominator 3, and the vehicle resumed highway speeds. Due to debris falling outside the tornado circulation, safety concerns necessitated bringing the ground station back inside the vehicle for periods.

     The ground station lost communication with the probe at a range of 14.5 km; however, later recovery of the probe and stored
data demonstrated that the probe was still transmitting throughout the entire flight. The authors determined that the shorter-than-expected radio frequency range was due to the proximity of the 16 gauge ballistic steel shell of Dominator 3 and was caused by obstruction of the signal and the Faraday cage behavior.



Subsequently, an omnidirectional antenna has been mounted externally on chase vehicles, and the vehicle shell serves as a ground plane. This configuration has demonstrated a line of sight ground level range of 26 km and an airborne range of
over 100 km.

## 4.3 Temperature and Relative Humidity

Holes were punched into the nose cone compartment of the rocket to promote airflow across the temperature and relative humidity sensors during flight, as pseudo-Lagrangian drifting within the tornado limits the airflow. Additionally, with minimal airflow, the effect of the electrical self-heating from the radio transceiver needed to be modeled. Other considerations were that
the temperature within Dominator 3 before launch was higher than the external temperature and the probe's thermal mass was moved from inside the vehicle to outside for launch.

Due to operational and time constraints prior to launch, a thermal analysis of the enclosure was completed after recovery. Treatments for thermal response exist for radiosondes that exhibit similar concerns where slower ascent rates result in a lack of airflow past the sensors contained within (Mahesh et al., 1997; Miloshevich et al., 2004).
Controlled experiments (Fig. 12) were conducted to compute the thermal time constant ($\tau$) of the probe within the enclosure, the probe outside the enclosure, and the self-heating effect. A fan, handheld anemometer, deep freezer, and ice-bath calibrated thermocouple were used.

These experiments identified constants for thermal response ($\tau$) and self-heating (S) to correct the temperature profile as measured by the probe. Additionally, the exact temperature was known at two points in time; launch (due to the calibrated
weather station on the roof of Dominator 3) and at a point in the descent where the air temperature external to the enclosure of the probe was equal to the measured temperature within the enclosure of the probe, and therefore no heat transfer was occurring.

Equation (1) was derived to correct the thermal data based on Newton's Law of Cooling (Miloshevich et al., 2004; Mahesh et al., 1997; Nash, 2015; World Meteorological Organization, 2017).
Air flow around the probe (v) was unknown but near zero as the payload moved as a pseudo-Lagrangian drifter. A meteorological balloon sounding (Topeka 00Z) was used as a reference for the environmental temperature profile by mapping the interpolated altitude in the sounding to the GPS altitude recorded by the sensor. It was found that there was a minimal range for plausible values for (v) that did not cause the corrected temperature to overshoot (v=0.07 m s$^{-1}$) the Topeka profile or undershoot (v=0.27 m s$^{-1}$). A midpoint value for (v) was chosen of 0.17 m s$^{-1}$ (Fig. 13), which provided the best curve fit
between the corrected temperature and Topeka profiles.

$$T_{env} = \frac{(T_t - S) - (T_0 - S)\,e^{-\Delta t/\tau}}{1 - e^{-\Delta t/\tau}} \tag{1}$$

where:

$\tau$ = time constant [Eq. (2)] ),     $T_t$ = sensor temperature at time t (°C),





**Figure 12.** Measuring the thermal time constant after removal from a deep freezer with a linear fan.

$T_{env}$ = temperature of the environment (°C),  $T_0$ = sensor temperature at time 0 (°C),

$\Delta t$ = delta time $t_t - t_0$ (seconds),  $S$ = self heating (4.18 °C)

$$\tau = \tau_0 \left(\rho v\right)^{-n} \tag{2}$$

where:

$\tau_0$ = time constant at zero air velocity (seconds),  $\rho$ = air density [Eq. (3)] ,

$v$ = air velocity in relation to the sensor (m s$^{-1}$),

$n$ = constant ranging from 0.4 (laminar air) to 0.8 (turbulent air); 0.8 is used

$$\rho = \frac{p}{RT} \tag{3}$$




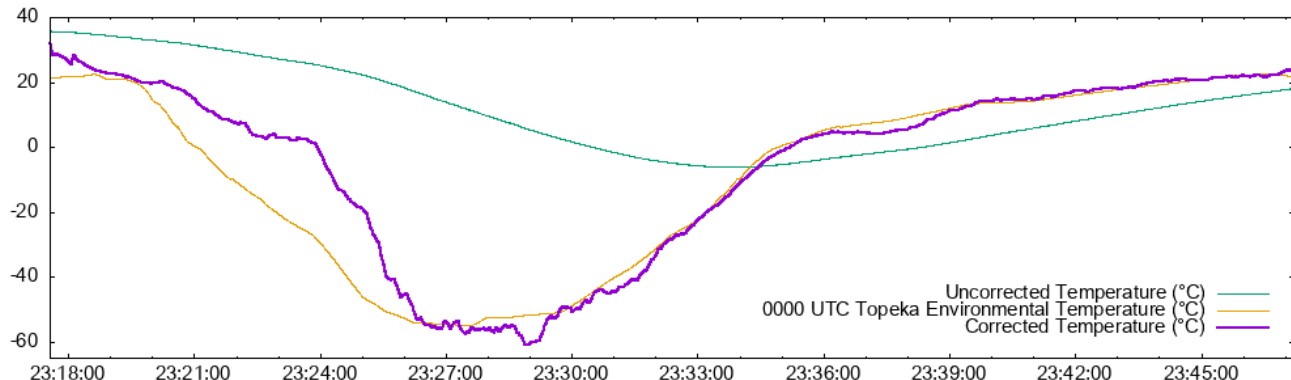

**Figure 13.** Thermal correction comparison showing uncorrected temperature (°C), corrected temperature (°C), and the 00Z Topeka sounding environmental temperature profile (°C).

where:

$p$ = sensor pressure (pascals),    $R$ = 287.058 (gas constant for dry air),

$T$ = 00Z Topeka sounding temperature (°K)


Due to the complexities and cost (> \$15,000) of the experimental setup to categorize the response for relative humidity in the enclosure, the authors decided to forego response time correction for this data; consequently, this necessitated using temperature instead of virtual temperature for the ideal gas equation [Eq. (3)]. This impact was deemed negligible for this study as condensing moisture was not encountered by the sensor, and design improvements had already been completed for
future deployments negating the requirement for thermal response correction. Additionally, this data component could be categorized later if required and budget permitted.

The design has since been improved to remove the requirement for the above data treatment by mounting the temperature and relative humidity sensors on a small thermally isolated daughterboard on the exterior of the rocket nose cone.

The effect of direct solar radiation (Wang et al., 2013) on the thermal correction was considered; however, as most of the
flight was within cloud and the sensor was shielded from direct solar radiation by the enclosure, it was concluded that any impacts from this would be negligible. To address this, future design improvements use two identical external temperature sensors, one black and one gold.

## 5  Summary and Conclusions

The authors launched a lightweight real-time trackable meteorological probe by model rocket into an EF4 tornado. Kinematic
and thermodynamic data were obtained from inside a tornado and associated mesocyclone. The high velocity of the model rocket successfully passed through any sinking air between the launch position and the tornado, and the probe was deployed by





the parachute and pulled inside the tornado. Three-dimensional direct measurements of wind, pressure, and temperature within a tornado, including updraft, had yet to be accomplished until the present study.

The ground-based launch of a rocket into the inflow region of a supercell proved effective in deploying a parachuted probe
into the tornado while avoiding the largest hydrometeors. The pseudo-Lagrangian trajectory of the probe allowed for the recording of three-dimensional wind speed through high-resolution 5 Hz GPS data.

While thermodynamic gradients are difficult to diagnose from the data of a single pseudo-Lagrangian trajectory, the success of the present study in capturing direct measurements of wind, pressure, and temperature inside a tornado will set the stage for future deployments of multiple simultaneous probes.

Multiple pseudo-Lagrangian probes collecting high-resolution meteorological data inside a tornado would allow a better understanding of pressure gradient and buoyancy forces. These forces drive the winds and behavior of tornadoes and supercells (Markowski et al., 2018; Bartos et al., 2022).

The probe transmitted high-resolution 1 Hz data in real-time from inside the tornado and provided 10 Hz (5 Hz GPS) data upon recovery. A flight time of 30 minutes sampled high three-dimensional wind speeds of up to 85.1 m s$^{-1}$, an updraft
velocity of 65.0 m s$^{-1}$, and reached the upper troposphere at 11,914 m MSL before descending. The probe completed 1.5 revolutions around the tornado in near axisymmetric flow during the 3 minutes within the tornado with a maximum updraft velocity of 65.0 m s$^{-1}$. The substantial pressure deficit during the tornado phase of the probe trajectory and the lack-there-of during the mesocyclone is expected due to tornadic winds being driven by tornado-scale gradients in pressure.

The 65.0 m s$^{-1}$ updraft velocity is the first direct measurement of vertical wind from inside a tornado. A maximum pressure
deficit of -113.5 hPa was recorded at an altitude of 475 m MSL within the tornado and dropped to less than -20 hPa by an altitude of 3,760 m MSL in the mesocyclone. The most significant pressure deficit being located just above the ground in this tornado would excite strong vertical velocities like those measured (65.0 m s$^{-1}$). Additionally, the maximum GPS three-dimensional speed of 85.1 m s$^{-1}$ at an altitude of 858 m MSL concurs with the surface wind of 76 m s$^{-1}$ estimated by the NWS damage survey; the trajectory within the tornado was near-circular, and the diameter of the probe's flight was in agreement
with the damage survey of 1.6 km.

The sharp transition of the probe trajectory over 10 s from a closed rotational path to pure updraft with negligible curvature starting at 23:19:40 UTC and an altitude of 1,128.7 m MSL (Fig. 14F), along with a significant pressure deficit of -113.3 hPa (Fig. 14G), is supportive of the probe sampling the RMW and the core of the tornado. The cessation of the rotational path of the probe is determined by the termination of the 90° phase difference between the u and v GPS sinusoidal velocity
components that indicate circular rotation together with a sudden increase in the updraft speed. The pressure deficit was still significant (-66.7 hPa) when the maximum updraft of 65.0 m s$^{-1}$ was recorded at an altitude of 2,171.3 m MSL, with near-zero curvature evident in the probe trajectory.

Previous studies involving radar observations, real ground measurements of wind and pressure, laboratory experiments, and computer simulations have shown that the inner core of a tornado inside of the core flow/RMW is comprised of a consistently
large pressure deficit with relatively minimal pressure variation within the core (Snow et al., 1980; Lewellen et al., 1997, 2000; Karstens et al., 2010; Wurman et al., 2013). The high-resolution modeling of tornadoes (Lewellen et al., 1997, 2000) shows a





**Figure 14.** GPS data and pressure perturbation sensor data for tornado and updraft portion of the probe flight.

relatively consistent pressure deficit within the tornado core for both one and two-celled vortices. A secondary maximum in pressure deficit as the rotational path of the probe abruptly ended is supportive of the transition of the probe from the RMW to the tornado's inner core. Therefore, the miniaturized pseudo-Lagrangian drifter effectively sampled meteorological data within the core flow and into the low-pressure core of a strong tornado, despite the extreme conditions and flying debris that typify core flow.

After 1.5 revolutions around the tornado, the probe underwent a consistently tilted, ascending path from an altitude of 4,300 m MSL up to the apex of 11,914 m MSL with a maximum GPS velocity of 83 m s$^{-1}$. The high ascent velocity, combined with the trajectory's lack of rotation and relative linearity, indicates that the probe was entrained very close to the center of



the mesocyclonic updraft. The authors reached this conclusion as a rotating mesocyclone flow was not indicated in the data as evidenced by the GPS heading and, therefore, lack of probe rotation around a center.

Improvements to the design since this experiment have included temperature and relative humidity sensors external to the rocket nose cone, a high dynamics GPS, and an omnidirectional rooftop antenna on the chase vehicle. Future work related to this project will involve simultaneously launching multiple probes from multiple chase vehicles. The authors believe this will

provide more insight into the three-dimensional structure and processes within tornadoes.

*Data availability.* The data received from the probe has been made available under a Creative Commons CC-By Attribution 4.0 International license: https://www.doi.org/10.17605/OSF.IO/Z64FD (Timmer et al., 2022)

*Video supplement.* A video animation of the probe trajectory (Supplemental1.mov) and a video recording of the rocket launch (Supplemental2.mov) has been made available under a Creative Commons CC-By Attribution 4.0 International license: https://www.doi.org/10.17605/

OSF.IO/UBTN5

*Author contributions.* RT conceptualized the project, managed the field work, and produced the rough draft of the manuscript; MS designed and built the probes, wrote the firmware, software, tested and curated and analyzed the data; MS prepared the manuscript in conjunction with RT and contributions from CB; CB designed and constructed the rockets, launch system, and maintained Dominator 3; SS drove Dominator 3 and provided safety monitoring.

*Competing interests.* The authors declare that they have no conflict of interest.

*Acknowledgements.* This Work was partly supported by individual Facebook subscribers, for which we are grateful. We are indebted to David Horsley for his review, suggestions, and helpful comments.

Additionally, the authors are grateful to Aaron Jayjack for weather spotting and safety monitoring during the deployment and to Matthew DuBois and Jeremy Belk from Leavenworth, KS, who assisted in locating and recovering the probe.

The authors appreciate the time, effort, and feedback provided by the peer reviewers of this manuscript.



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
