# Peer review of "Design and Rocket Deployment of a Trackable Pseudo-Lagrangian Drifter based Meteorological Probe into the Lawrence/Linwood EF4 Tornado and Mesocyclone on 28 May 2019"

_EGUsphere, 2023_

## Author Response (AR1)

**Reply to Anonymous Referee #3**

The authors appreciate the time and effort that Anonymous Referee #3 has dedicated to providing extensive and valuable feedback on our manuscript to improve the work. We are grateful for your insightful comments on our paper.

Overall comments:

1). There are numerous instances throughout the manuscript where paragraphs are single sentences. I would encourage the authors to consolidate similar themed statements into single coherent paragraphs rather than breaking them apart into multiple paragraphs. This may just be an artifact of the preprint, but it should be addressed either way.

We thank the reviewer, and this has been corrected.

2). The current study describes the observations collected by the probe and compares it with a single radar scan (KTWX, 0.7 degree elevation). While this is important, the radar scan takes places before the launch of the rocket and there is no additional analysis presented. I think the study would benefit considerably by additional radar observations throughout the nearly 30 min flight time of the probe, as well as exploring higher elevation angles to provide deeper context to the observations. While the observations themselves are novel, placing them into the context of the larger storm enhances the scientific value and allows for future work to build off of. I encourage the authors to expand the analysis and discussion section by utilizing additional radar data.

Figure 8 includes a radar reflectivity and velocity image from as close to the launch time as possible, and was included in the manuscript for the purpose of showing the ideal launch location relative to a high-precipitation tornadic supercell (inflow notch just northeast of the tornado). This figure was not included for comparison with the data collected by the probe during flight as stated by the reviewer. The KTWX radar site is located about 50 km northwest of the launch location and was even farther away from this tornado, which progressed to the northeast of the launch location south of Lawrence, KS. Hence, only the low-level mesocyclone can be resolved with the smaller-scale tornado cyclone falling within the radar resolution at this range. While we agree that an additional figure with multiple radar images with the probe GPS location superimposed would provide even more meaning and meteorological context for the observations collected, this publication is more focused on the technology and deployment methodology.

Major comments:

1). Line 24-26: the final statement of this paragraph is slightly abrupt and out of place given that the manuscript has not mentioned the probes in the body of the text yet (only in the abstract), and requires a bit of a rework. This should be moved to the end

of the introduction section after setting up the "problem", which is a lack of direct observations inside the tornado vortex. A final paragraph such as the following would suffice:

"Given the above mentioned studies, it is clear that there is a notable lack of direct observations inside or near the tornado vortex. To address this observational need, the present study describes a new, miniaturized, pseudo-lagrangian sensor deployed from a low-cost surface based model rocket that is capable of obtaining these observations inside the circulation. The design of this sensor as well as a case study showing its use is presented here."

With the above you could remove, or incorporate, the current final paragraph of the introduction with the above suggested paragraph. This change, along with other suggestions, would streamline the flow of the introduction sections and lead into the methodology.

The authors welcome the suggestion and have reorganized and amended the introduction as per the reviewers suggestions, resulting in a less abrupt and improved flow.

2). Lines 68-74, this is great information but belongs in the discussion of the actual sensor. This should be moved to section 2 under methodology. This is simply a major comment as it involves restructuring the paper to a degree.

This has been moved to the Methodology section as per the reviewer's suggestion.

3). I have numerous questions and concerns regarding the probe design itself and the ability of the probe to accurately represent the (thermodynamic) environment in which it is travelling. I will list each here.

- The authors state on line 121 that a conformal coating was added to the probe in order to protect the sensors from damage. This however is adding a barrier between the pressure, temperature, and humidity sensors and would effectively isolate them from the ambient environment. The RH sensor for example is designed to measure water vapor molecules attaching and detaching from the sensor. The conformal coating on the senor board is specifically designed to prevent that process from happening, thus these two do not work together. Coatings on a temperature sensor can reduce the connection to the ambient environment and slow down the response to changes. And depending on the thickness and uniformity of the coating it could completely seal the pressure sensor. There is no supporting information in the manuscript as to what effect this has on the performance of the sensor. Can the authors comment on the changes in response to the probe this conformal coating has? Are there tests that were done to determine the accuracy/responsiveness of the probe that were completed before and after the conformal coating was applied to showcase the effect the coating has, or prove that it has none?

A silicone conformal coating was manually applied to the electronics (L121), but not all the sensors for exactly the reason the reviewer has highlighted and for other reasons. It is mentioned that a PTFE barrier (L121) was applied over the port to the BME280. The 2 sensors that were not conformally coated on their sensing areas were the BME280 (Temperature/Pressure/Humidity) and the SI7053 (Temperature) sensors - the remainder of the sensors would not be impacted by the coating (IMU for example). The case for not conformal coating the BME280 was due to it having a small hole (port) that needed to be open for airflow for sensing as well as the concerns highlighted by the reviewer. Additionally, the MEMS sensor RH polymer is sensitive to the volatile chemicals present in conformal coating, soldering and washing. The port was protected with an impervious tape disk during coating and washing, with the coating only being applied around the base, not on the top/port. Additionally, the SI7053 was protected on its sensing surface via a tape rectangle. The tape was removed after the assembly was completed. i.e. Neither of the 2 sensors were conformally coated over their sensing areas, and were also protected from accidental coating and slumping of that coating.

Specifically the BME280; due to the port design can be damaged by water ingress into the port after assembly, so a permeable PTFE barrier (L121) was applied over the port to allow airflow to the internal MEMS sensor (the BME280 was not available with a PTFE barrier). This approach is standard practice for MEMS sensors involving a Relative Humidity component, for example:
     https://sensirion.com/products/catalog/SHT45-AD1F
the reason for this is that PTFE membranes allow water vapor and air transmission but not water (rain, condensing moisture) without noticeably impacting the response time for temperature or humidity.

To illustrate this, comparing a similar sensor both with and without the membrane, e.g.:

https://sensirion.com/products/catalog/SHT45-AD1F
https://sensirion.com/products/catalog/SHT45/

it can be seen that the RH response time is 4s on both, and the temperature response time is 2s, i.e. impact of the PTFE membrane is negligible.

We tested two PTFE membranes for the BME280: white and pink plumbers tape. The white plumbers tape was the same thickness as the membranes typically used for RH sensor protection. The much thicker pink tape we found did impact the response time slightly, but we found in testing that the white tape had no detectable impact to temperature, humidity, or pressure response times, which fits with the predicted usage of a PTFE membrane of that thickness.

We agree that although the explanation in the manuscript, alludes to special treatment for the BME280 in particular, it's not explicitly stated, therefore we have amended this to:

"The electronics were waterproofed using a silicone conformal coating to protect from unpredictable behavior due to current leakage attributable to moisture.  The BME280 and SI7053 sensors were protected from this coating during assembly to avoid damaging the BME280 polymer due to volatile chemicals and a reduction of the thermal response on both sensors due to the insulation of the coating. A permeable polytetrafluoroethylene (PTFE) barrier was placed over the port on the BME280 to avoid damage from water and condensing moisture during flight as is typically applied to Relative Humidity polymer-based MEMS sensors."

- There is no presented documentation of any testing done to evaluate the performance of the probe. The first data from the probe shown is in flight during the actual deployment. Given that this is an entirely new system, detailed testing must be done prior to use to ensure that the design is functional and accurate for the intended purposes. Figure 12 and some text around lines 280 suggest that experiments were done to quantify the response of the sensor, but the data from those experiments is not shown in this manuscript nor is information provided regarding the response time of the system (this does not mean the response time of the sensor listed in Table 1, but rather the complete "as deployed" system) or any description of the outcome of those tests if they were completed. This is a significant omission as the sensor package must first be proven reliable before experimental data can be trusted. Additionally, this discussion should be in Section 2 when introducing the details of the probe and not in Section 3 during the case study.

The authors should add a section on the testing done on the original sensor (not the modified one discussed at various points) along with supporting data to demonstrate the performance of the system. What is the response time of the actual system, with the conformal coating and placed inside the nose cone? How does this response time vary with aspiration rate over the external nose cone to simulate the expected fall speed? How were experiments set up to determine the responsiveness of the probe? What are the limitations of the probe design and where is it likely to experience problems? How does this probe compare to currently available instruments like a radiosonde or even surface-based sensors during general conditions and during rapidly changing environments? What tests were done to ensure that the probe was able to accurately observe conditions over the expected range of values for pressure, temperature, and relative humidity?

Testing was completed on the recovered sensor package as flown, the results from the later modified ones were not used in the manuscript.  A detailed document discussing the Thermal Lag Treatment was written shortly afterwards as we completed our analysis.  However in the interests of brevity/clarity for the manuscript, only pertinent parts were included in the manuscript due to length concerns.

In light of the reviewers comments, we have uploaded the document to permanent storage located with the flight data.  Additionally, we have cited this document in the

manuscript for further information when discussing the "Controlled experiments".  This document details how the probe's sensors were tested and how the thermal lag was corrected.

https://doi.org/10.17605/OSF.IO/BQ93T (ThermalLagTreatment.pdf)
(the remaining files at that location are graphs and data from the tests according to the filename title)

The fall speed of the probe is unrelated to the response rate other than by scaling, as the aspiration rate over the sensor controls the response rate.  When we derived the treatment formula and applied it, we found that, there was a limited range of flow rates past the sensor that resulted in plausible values.  For the definition of plausible, we used the Topeka sounding as a base reference.

Despite the tornadic environment not being identical to the Topeka sounding, it had to be a similar fit to the curve, especially on descent, as the distance between the two was never more than a few degrees Celsius.  This approach negated the need for accurate airflow measurement past the sensors, which is difficult to attain at such low speeds and small spaces.

As velocities produce an implausible solution with opposite extremes at V=0.07m/s and v=0.27m/s, we chose a midpoint at 0.17m/s which provided a good match where it didn't overshoot or undershoot.  Although referencing a nearby sounding in a different environment was less than ideal, we felt it was a valid approach due to the limited range of the solution as we were solving for only one variable (velocity past the sensor).

Considering that an RM Young Propeller Anemometer (05305) only begins to measure wind above 0.4m/s with a +/- 0.2m/s accuracy and an RM Young Ultrasonic Anemometer (86000) has a +/- 0.1m/s accuracy, we felt that in this case the accuracy of the model and the subsequent curve fitting was far better than it was possible to measure.

Comparatively, the sensors used in the probe are of higher quality and accuracy than is used typically in radiosondes, as the technology is newer.  Additionally the response rates and accuracy are specified in the datasheets; these were evaluated during bench testing to confirm that they were within range.  Relative humidity data was not included in the manuscript due to the inability to fit a valid model without specialized equipment outside of our budget and so was excluded from the manuscript, however a future design change solved this for future deployments.

- From the discussion it appears that the probe collects data from inside the nose cone after separating from the rocket, with a parachute to allow the pseudo-lagrangian behavior, and a few small holes in the nose cone to presumably allow airflow into the nose cone. Furthermore the probe is housed in the nose cone with "expanded polystyrene foam for thermal protection" (line 148) surrounding it. The combination of these factors would lead to a very isolated, poorly mixed

connection to the ambient environment with little to no flow over the sensors, effectively disconnecting it from the ambient environment (this is clearly evident in figure 13, more on this in major comment 7). It is also unclear from the manuscript in what orientation the probe collects data in (see major comment 4). Is the nose cone vertical or upside down during data collection? What testing was done with the probe inside the nose cone and outside to show any differences in the responsiveness to changes in the environment? What assumptions are being made regarding the "fall" velocity of the nose cone once the parachute is deployed? What direction is airflow in theory supposed to be entering the nose cone and where does it exit? Were there any tests or calculations performed to determine the flow rate inside the nose cone near where the sensors are located? Were there tests done on the addition of the polystyrene foam to determine what effect this has on the responsiveness of the probe? What design iterations were considered here as alternatives to this design (eg. what is the history of trials that lead to this)?

A foam plug was positioned at both ends of the sensor to prevent movement of the sensor in flight in the larger payload bay and also for thermal protection of the lithium ion battery.  The area around the board sensors was not covered with foam, and the venting was around this area to promote airflow.

Airflow past the sensor was less than expected, but despite that we feel it was still connected to the ambient environment albeit with high thermal lag.  From a verification perspective, this is demonstrated by the curve match detailed in the thermal lag treatment above.

The manuscript has been updated to read "The probe was housed within the nose cone compartment with expanded polystyrene foam plugs at both ends to constrain the movement of the probe within the enclosure during flight and for thermal protection of the battery, whilst still permitting airflow across the sensors. "

The nose cone is vertical during data collection with pointed end towards the ground, however we do not expect the orientation to produce different measurements or changes in airflow due to the ventilation holes being in a horizontal circular ring. Figure 3 has been added to demonstrate the probe orientation during pseudo-Lagrangian flight.

Testing was completed both with and without airflow, in addition to the bare sensor and in the rocket cone as detailed in the Thermal Lag Treatment document detailed previously. Also, we have uploaded response curves for this testing and referenced it in the manuscript:

https://doi.org/10.17605/OSF.IO/BQ93T

The airflow past the sensor is a more direct modeling than the airflow past the enclosure, as the thermal lag was the response of the sensor to the airflow past it, not

past the enclosure (descent speed).  There were a number of holes drilled in the nose cone, a main ring and additional holes at the opposite end of the sensor to promote airflow.  Airflow could enter from a number of directions depending on the movement of the probe, so a sensor centric approach was used with holes at both ends.

The testing was completed with the polystyrene foam in place to negate the impact of the polystyrene foam impacting the results.  The design was iterated after this deployment, due to the limited tornado season, short development time and lessons learnt from deploying into a tornadic environment.

Testing prior to deployment consisted of rocket tests and design refinements related to deployment and pressure measurement, but as there was no supercell during testing and it wasn't possible to test these sensors to the stratosphere, the probe would descend back down and had limited range in height.  Since then, the probe has also been converted into a balloon deployment package and has been additionally tested in both pre-tornadic and non-severe weather environments with improved response times, negating the need for the Thermal Lag Treatment on future deployments.

- Given that the probe is located inside the nose cone, with the pressure sensor mounted to the circuit board itself, there are potential local pressure perturbations due to the design that can negatively affect pressure observations. As air flows over and through objects, Bernoulli effects can lead to potentially significant pressure perturbations depending on flow rates over the objects. It is for this reason that pressure observations at the surface are taken using static pressure ports. Are there tests done that show pressure observations both in and out of the nose cone at varying speeds over an expected range to document any errors the nose cone may be imparting on the observations? There are relatively simple tests that could be performed to document this.

Due to the low airflow (0.17m/s past the sensor and 4-6m/s past the enclosure) and the probe moving in pseudo-Lagrangian fashion it was determined that the Bernoulli Principle effect was negligible and negated the need for a static pressure port during the Lagrangian phase of flight.  The initial rocket phase of flight was at higher speed, but the limited usefulness of data during that phase of flight meant that the extra weight of a pressure port wasn't required as that part of the data had limited use.  Additionally the space around the sensor inside the probe enclosure acted as a crude pressure port anyway.

- As it stands currently, I have serious concerns about the validity of any data collected by the probe, especially without documentation. Given these concerns, my recommendation is for the authors to entirely remove any discussion and presentation of data from the thermodynamics of the probe (RH isn't discussed currently, only temperature and pressure). The kinematic data is novel and worthy of publication but I am not confident in the ability of the probe to accurately

represent the thermodynamic environment. If the authors wish to keep the discussion of temperature in the manuscript, then significant additions must be made to the manuscript in order to justify and prove its usefulness, with the above questions addressed in the section.

We hope that additional references added on the Thermal Lag Treatment are sufficient to address the reviewers concerns. As has been demonstrated, adding the the full details of this testing that the reviewer required, significantly lengthens the manuscript and detracts from the main purpose of the manuscript, so we feel that adding it as a reference is a good solution to both provide the information and avoid the detraction.

4). It is unclear from figure 2 how the sensor board actually flies during data collection. The separation charge after burnout from the rocket motors separates the nose cone (which contains the sensor board) from the body, but is the parachute directly tied to the sensor board alone (I think this is true but it's unclear). More details are needed as to the actual mechanism of deployment for the sensor board, and could benefit from an additional figure or diagram depicting the separation of the sensor board from the rocket body/nose cone. The orientation of the system during data collection is critically important to evaluating the data it has collected.

The sensor resides in the rocket payload bay and the payload bay is attached to the parachute after separation. Figure 3. has been added to clarify the various stages of flight and the parts involved as per the reviewers suggestion.

5). The authors make the point that the probe design is in part to obtain observations without the need to directly enter the circulation (Lines 10-11). I think this is an important point and presents an opportunity for a discussion on deployment/operational safety. I would strongly encourage the authors to add a section at the end of Section 2 (could call it 2.5 Deployment Safety) where safety considerations and a planned deployment strategy are laid out. This would hopefully discourage others from attempting to recklessly or thoughtlessly recreate these observations and would demonstrate the authors commitments to maintaining safety. It's a good conversation to have, and this is a good opportunity to have it.

We thank the reviewer for their suggestion, and section (2.5 Deployment Safety) has been added to address this.

6). Lines 250-254, in this paragraph it is unclear whether data in the plots and associated discussion is being presented during the rocket flight and before the probe separation. If the plots are showing data only AFTER the separation, then a note to this effect should be placed in the text. If the plots include data during the rocket phase, this should be clearly indicated in the text and on the plots. The supplementary data showing the video animation of the trajectory appears to indicate the rocket path (in red) before the deployed parachute phase. Given the relatively short rocket phase it is my assumption that this discussion and figure show only deployed data, but it should be made more clear.

The data includes the flight. For the GPS data however, that is essentially 2 points at the beginning (launch and signal reacquisition).  This has been clarified with the following statements added to the manuscript:

GPS:
"and the data and figures include this phase of the flight."

Pressure:
"It should be noted that during the launch of the rocket prior to pseudo-Lagrangian flight that the pressure was impacted by Bernoulli's Principle due to the high velocity of the rocket flight, the data and figures include this section of the flight."

7). In connection with major comment 3, Figure 13 shows temperature data from the time of launch (or separation from the rocket, this detail is unclear) from the probe in comparison with data from the 00Z Topeka sounding.

The temperature data is shown from the time of launch. This is required as the thermal lag correction model requires contiguous data from the start of the recording to avoid any discontinuity.

Added "from launch" to the plot caption for Fig 13.

- As a first point, comparing these two soundings is problematic as the probe was launched into convection and the Topeka sounding presumably was not, and was nearly 50 mi away. These environments are not the same and forcing the probe to match the Topeka sounding as what is being shown in Figure 13 completely removes the point of launching the probe into the tornado in the first place. This is effectively forcing the rocket probe to match the Topeka sounding, in which case why not just look at the Topeka sounding?

  The Topeka sounding was used to represent the ambient environment outside of the supercell storm. This was particularly important for the derivation of the pressure perturbations measured inside the tornado and mesocyclone by the probe (Fig. 14). The location of the Topeka sounding outside of convection was preferable to use as a reference for the temperature correction in this case, since it is used to represent the ambient environment outside of the supercell and tornado. The expected temperature at a given pressure level in the ambient environment outside of the tornadic supercell was achieved with the Topeka sounding, which was even located on the same side as the outflow boundary draped across northeast Kansas that day. We considered removing the temperature observations from the paper, but believe they are important to show these probes are capable of measuring temperature inside the tornado. We have moved the sensors to outside of the enclosure for future launches, which would ideally remove the need for substantial temperature correction.

- A larger issue with this figure is that the sensor probe (pre-correction) is clearly disconnected from the ambient environment. The entire flight shows a temperature range from roughly 35 C to only as low as -10 C despite reaching altitudes as high as 11km, and follows an extremely smooth decay response. Meanwhile the Topeka sounding shows a general environmental change from roughly 20 C to almost -60 C. (again taking into consideration that these are not in the same environment) Assuming the data points are matched by altitude in this figure, at its highest point in altitude the probe indicates a temperature of +15 C while the Topeka sounding indicates the temperature should have been closer to -80 C. The general look of the temperature trace is a clear indication that the probe is almost completely isolated from the ambient environment and has an excessively long response time (likely due to the factors I list in comment 3), thereby making the temperature observations unusable. Can the authors comment here on why they think the data are scientifically valid given the excessively large disparity between the probe temperature and a generic environmental profile in the region?

The corrected temperature follows the Topeka temperature profile and the corrected temperature is a lag correction to the original lagged (uncorrected temperature). Treatments for thermal lag are used, as detailed in the literature for Radiosondes ((Mahesh et al., 1997; Miloshevich et al., 2004).  In this case, the lag is more extreme than typical, as per previous discussion, but still a conductive process so:

1. Environmental Temperature = Sensor Temperature = No Heat Transfer
2. Environmental Temperature < Sensor Temperature = Transfer To Environment
3. Environmental Temperature > Sensor Temperature = Transfer To Sensor

We agree that the response time is long, however the lag is a phase delay, and the delay can be computed (time constant), so the original time series can be still be recovered.

In a stable environment the sensors ascent should follow the adiabatic lapse rate closely. Given the highly unstable environment as shown by the velocity and accelerometer data during the ascent, it was concluded that conditions would be closer to the adiabatic lapse rate during the later part of the descent phase when the flight was more stable, than the ascent phase where conditions were unstable.   The Topeka 00:00Z sounding was used to serve as a stable reference sounding and to validate the adjustments applied.

The case for the validity of this data treatment can be seen by the reversal of the process:

 The time decay formula:

$$V(t) = V_0 e^{-t/\tau}$$

can be reversed to the original series provided Tau, V0 and t are known.  V0 is known and is the first temperature data point as is t.  Tau (the Time Constant) is unknown, however as Tau is adjusted within that equation from the correct value, the result changes from plausible to implausible.  Given the assumption that the sensors descent should follow the adiabatic lapse rate, and that the Topeka 00:00Z sounding is a good approximation to that, the range of values for Tau are very limited.  Tau is essentially the airflow rate past the sensor, i.e. more airflow equates to a shorter time constant. Tau is corrected for air density as the probe can be located at different altitudes.  However the probe is still always descending due to gravity in relation to the air mass, even if the air mass and probe are ascending relative to the ground,

- It is curious how the "corrected" data have a significant amount more variability than the original uncorrected data. Generally, corrections tend to work the opposite way and smooth the data further. This raises questions and concerns about the accuracy of the correction that was applied as it appears to be unstable and variable, changing from point to point. Additionally, the logic behind the flow rate being near zero due to the sensor being pseudo-lagrangian is flawed. Given that the sensor/nose cone has a mass, it has a fall speed in all Even in updrafts, the probe would still be "falling" relative to the environment around it, even if it's ground relative velocity is positive. The velocity inside the nose cone where the probe is however is probably zero or near zero given the behavior of the temperature response. Can the authors expand on their correction factor, explaining how the corrected data has significantly more variation than the original data? Can the authors comment on the applicability of a correction factor that results in a nearly 80 C swing in temperature? Can the authors comment on using the Topeka sounding to force the data to a specific profile rather than determining the correction factor organically off known and controlled test data?

The corrected data has more variability compared to the uncorrected data due to the uncorrected temperature exhibiting thermal lag which smoothes the time series due to the slower rate of heat flow.  i.e. highly variable temperature translates to less variable thermally lagged data, and reversing that process results in the original highly variable temperature data.  Additionally, the probe was in turbulent air in the tornado and mesocyclone resulting in pressure, altitude and temperature variability from sample to sample. The sample rate at 10Hz was frequent (which is unusual compared to Radiosondes for example which typically sample at 1-6s per sample), meaning that this turbulent environment is being sampled effectively by the 10Hz sample rate. Comparatively, the variability is considerably less on the descent phase after 23:31:00 as can be seen in Fig 13. (now Fig 14), illustrating that the variability was largely a factor of the environment in which the probe was embedded.

We agree that despite the descent rate being 4-6m/s as specified in the manuscript, that the airflow past the sensor inside the probe enclosure is close to 0.17m/s by the calculations/modeling provided.  The correction factor is now explained more fully in the

Thermal Lag Treatment pdf as detailed previously:  https://doi.org/10.17605/OSF.IO/BQ93T

The 80C swing in temperature is in close agreement with the Topeka sounding, so indicates that the correction factor is representative.  The discussion about the "forcing" to the Topeka sounding we have detailed above.

- "Corrections" are not meant to modify the data to this extent and are not appropriate in this context. Given the apparent behavior of the sensor in regards to temperature, I suggest removing the thermodynamic observations from the manuscript entirely, unless the authors can demonstrate that the probe is capable of collecting accurate and representative observations through rigorous ground tests.

We hope that additional references added on the Thermal Lag Treatment are sufficient to address the reviewers concerns.  The authors feel the treatment is valid given the very limited airflow velocity range that provides plausible solutions to the model.  We posit that a close fit to the Topeka sounding with limited viability outside the limited variable range is evidence that the approach used is valid.  Additionally we feel that experimental measurement of the airflow pass the sensor would be difficult to achieve to the level accuracy required at such low speeds and within a limited space, and ultimately the validation would need to be against an actual temperature profile (of which the closest is Topeka). Taking those considerations into account, the direct approach of fitting to the overall curvature of the Topeka sounding, especially the descent phase, skips a likely inaccurate measurement and directly targets the end validation.

Minor comments:

Except where detailed below, all minor comments not listed below have been addressed as per the reviewers' recommendations:

Line 4: Suggest adding "horizontal" before "….velocity of 85.1 ms-1…" to clarify direction of motion

This has been changed to "three-dimensional speed" to match other locations in the manuscript where this value is discussed. Although speed being scalar would not generally have a direction, and the velocity vector is changing constantly, we feel that it's important to still illustrate direction as highlighted by the reviewer despite speed being the correct term to use in this instance. Therefore we have qualified speed with a direction.

Line 4-5: The current statement indicates a measured pressure deficit of -113.5 hPa at an altitude of 475 m. Is this pressure deficit relative to the ambient pressure at that

altitude or relative to the surface pressure? If it's surface pressure, is this taking into account the vertical decrease in pressure due to altitude (standard atmosphere is roughly 1 hPa per 10 m, which would mean 47.5 hPa of that change is just due to altitude, and there are questions whether a standard atmosphere applies in this context)? If it's relative to the ambient pressure outside the tornado vortex how is that ambient pressure obtained? Clarification of the context is needed here. A simple statement of the context is sufficient here, with further expansion in the body of the text.

The pressure deficit takes into account the expected pressure drop with altitude. i.e. It is the pressure deficit from the expected pressure at that altitude. It is referenced to the pressure on the surface at launch. The following equation is used for pressure altitude: https://www.weather.gov/media/epz/wxcalc/pressureAltitude.pdf. i.e. In the above example, the -113.5 hPa does not have 47.5hPa of pressure change attributed to the altitude difference in that value. It is relative to the ambient pressure at the launch position (which was outside the radius of maximum winds and outside of the circulation).

Added the words "altitude-corrected" to "pressure deficit of -113.5 hPa".
In the body of the text, added "launch-referenced and altitude-corrected"…

Line 6: please specify whether this velocity is vertical or horizontal

Clarified the 65.0 m s-1 with the word "speed". The manuscript has the word "tilted", for example "tilted ascent" to indicate vertical movement. Due to wind shear, the probe did not go straight up, so we do not feel comfortable specifying vertical in this case.

Line 8: was the probe tracked all the way to the surface or was the transmitted data cutoff at some altitude?

Transmission cutoff near the tropopause however, the flight data to the surface was recovered from the onboard storage when the sensor was recovered. This was detailed in the body of the text, but has now been clarified in the abstract.

Line 23-24: Are there references for the mentioned high-resolution cameras and drone videos? Even if they're social media posts I think it's worth citing the work.

The following references have been added:

Wakimoto, R. M., Atkins, N. T., Butler, K. M., Bluestein, H. B., Thiem, K., Snyder, J. C., Houser, J., Kosiba, K., and Wurman, J.: Aerial damage survey of the 2013 El Reno tornado combined with mobile radar data, Mon. Wea. Rev., 144, 1749–1776, 2016.

Groenemeijer, P.: Great footage. I tracked an object that seemed to be lifted particularly fast early in the video. I calculated speeds partly over 100 m/s (360 km/h or 225 mph).

Later, debris can be seen to move horizontally at similar speeds just 20ish meters above the ground., https://x.com/pgroenemeijer/status/1521027958827274240?s=20, 2022.

Line 135: This would be a good place to move the "previous attempts" information/ discussion.

With the changes the reviewer suggested for the Introduction which resulted in improved flow and the moving of the "previous attempts" to the end of the introduction, the authors feel that the "previous attempts" now are a better fit at the end of the introduction instead of the Rocket And Launcher Design section.

Line 162-167: these statements can be combined into a single paragraph, consider reducing text for redundancy (the ""in a cleared area, free of tall trees, power lines, buildings, and dry brush and grass" is repeated)

These have been combined into a single paragraph as suggested. In a previous peer-review, there were some minor differences in wording highlighted between the 2 sets of regulations which required clarification to be added to the manuscript, therefore we feel it necessary to highlight this from the two different sources.

Line 285: how is the ambient external temperature known at a point during the descent at the location of the probe?

The enclosure/sensors exhibited thermal lag, so the internal temperature is phase delayed. If the environmental temperature is less than the internal temperature, then the internal temperature will drop towards the environment temperature as heat is transferred to the environment, and conversely, if the environmental temperature is higher. Therefore there is a point of inflection on the descent where the slope is zero in this case (due to the thermal lag) when the environmental temperature crosses internal temperature, and at that point, the environmental temperature is known as there is zero heat transfer as both temperatures are identical. Due to the thermally lagged temperature only crossing the environmental temperature once, this is a reasonable conclusion as there is only one point of inflection in the uncorrected temperature (Fig 13).
To further clarify, the text has been amended to "and at a point in the descent where the air temperature external to the enclosure of the probe was equal to the measured temperature within the enclosure of the probe, and therefore no heat transfer was occurring to or from the sensor internal to the probe (Fig 13)"

Line 290: This statement cannot be true as the probe will always have a descent speed given that it is not "freely floating" in the atmosphere. Earlier in the manuscript the authors estimated the ambient fall speed with the parachute deployed to be in the 4-6 m s-1 range, that assumption should carry here.

We thank the reviewer for highlighting this typographical error.  Airflow around the probe is independent from airflow past the temperature sensor and the "v" refers to airflow past the sensor (0.17 m s -1) not the enclosure (4-6 m s-1) which was stated in the manuscript at L290.  Further on in the manuscript, it was stated correctly at L304 "v = air velocity in relation to the sensor (m s$^{-1}$)".

L290 has been changed to correct this error, and further clarification has been added:

"Although the descent rate of the payload was known when moving as a pseudo-Lagrangian drifter, the unknown airflow around the sensor (v) could be determined to lie within a range of possible values. The sensor airflow (v) must be less than the descent rate due to the enclosure having venting but not being completely open to the environment. "

**Reply to Anonymous Referee #2**

The authors appreciate the time and effort that Anonymous Referee #2 has dedicated to providing
valuable feedback on our manuscript to improve the work. We are grateful for your insightful comments on our paper.

Overall a good paper that accomplishes what it promises – it describes the rocket-launched drifter and presents results. Below are several minor comments:

1. The term "pseudo-Lagrangian drifter" is applied because the path of the drifter is assumed to be the path of the wind. The sink rate of the drifter could be determined and used to derive a better estimate of
the wind. Why was this not considered?

The authors considered correcting the "W" wind component for the sink rate of the pseudo-Lagrangian drifter. There are a number of factors that
impact the sink rate, including pressure perturbation within the tornado vortex and updraft, altitude due to air density, and varying sink rates within the downdraft,
all of which would be dependent on temperature too. Rather than adding a number of corrections to adjust for the sink rate which could also interact with the temperature and thermal lag correction or be co-dependent, we concluded that a more conservative approach was to specify the range of 4-6 m s-1 calculated for the air densities encountered rather than having the "W" component potentially higher than experienced by the probe.

2. The following paper also describes a pseudo-Lagrangian drifter designed to operate in severe storms. This reference did not deploy into a tornado, but does include results from the storm environment:
Sara Swenson, Brian Argrow, Eric Frew, Steve Borenstein, Jason Keeler. "Development and Deployment of Air-Launched Drifters from Small UAS." Sensors. 19(9): 2149, May 2019.

This reference has been added: "Using pseudo-Lagrangian drifters for infiltrating thunderstorms and supercells with sensors has been accomplished using a
dual-balloon approach (Markowski et al., 2018) and a single-balloon approach where the balloon is filled with helium during flight (Swenson et al., 2019)"

3. The paper would be improved by moving comments about future design elements into a new section on Lessons Learned / Future Work. The current style of describing improvements and new designs in the main body weakens the paper and downplays the significance of these results.

All references to future work/changes within the body of the text have been moved to the section "Lessons Learned/Future Work".

**Reply to Anonymous Referee #1**

The authors appreciate the time and effort that Anonymous Referee #1 has dedicated to providing valuable feedback on our manuscript to improve the work. We are grateful for your insightful comments on our paper.

Design and Rocket Deployment of a Trackable Pseudo-Lagrangian Drifter based Meteorological Probe into the Lawrence/Linwood EF4 Tornado and Mesocyclone on 28 May 2019

Author(s): Reed Timmer et al.

General comments

The manuscript presents the design and deployment of a meteorological probe for sampling the tornado flow. The study shows that lightweight meteorological probes launched by rockets can allow measurements of wind and thermodynamic conditions of the tornado flow. So far this type of measurements were very hard to obtain due to strong wind and strong pressure gradients associated with the tornado.

The paper is within the scope of the journal and is addressing a relevant scientific question regarding the measurement of wind and thermodynamics in tornadoes that will allow a better understanding of their impact. Thus, a novel method is presented to measure directly the three-dimensional wind and thermodynamics inside a tornado. The description of the meteorological probe and the rocket is complete allowing their reproduction by fellow researchers. Furthermore, the title and the abstract are clear and concise.

Specific comments

I think a short explanation regarding the pseudo-Lagrangian drifters will be helpful for the readers.

The following sentence has been added to the introduction "A pseudo-Lagrangian drifter is a sufficiently miniaturized, instrumented probe that is intended to move along with the wind flow, such that the speed of the probe is very close to the wind speed of the air in which it is embedded."

Lines 101-102: This is not very clear for a general reader as the sentence is quite technical: "Using quaternions from the IMU and the sensor-fusion, the Tait-Bryan angles for pitch, yaw, and roll (as is used in aircraft orientation) were derived".

Changed to: "Using the IMU sensor-fusion of the magnetometer, accelerometer, and gyroscope, pitch, yaw, and roll angles were derived."

The future design and sensors changes should be included in a separate sections ("Future work").

All references to future work/changes within the body of the text have been moved to the section "Lessons Learned/Future Work".

Technical corrections

Lines 47-52: This paragraph consists of one sentence. I think it could be integrated with the next two paragraphs.

Restructured from 3 paragraphs to 2, to read as follows:

"Over the past two decades, field experiments such as VORTEX, VORTEX2, and Project TORUS have improved the understanding of supercells, tornadoes, and particularly tornado environments using mobile Doppler radar (X-, W-, Ka-band), in situ ground-based probes, balloons, mobile mesonets, and Unmanned Aerial Systems (UAS) (Straka et al., 1996; Bluestein et al., 2003; Wakimoto et al., 2003; Bluestein et al., 2004; Samaras and Lee, 2004; Blair et al., 2008; Weiss and Schroeder, 2008; Karstens et al., 2010; Kosiba and Wurman, 2010, 2013; Wakimoto et al., 2011; Wurman et al., 2012; Tanamachi et al., 2013; Winn et al., 1999; Samaras, 2004; Pazmany et al., 2013; Frew et al., 2020; Houston et al., 2020; Markowski et al., 2018). However, the UAS technology of Project TORUS is not intended for direct measurements inside of a tornado core flow (Frew et al., 2020; Houston et al., 2020).

Mobile radars are mainly limited to measuring horizontal wind of storms and tornadoes, with the vertical component being inferred due to inclined measurements unless the measurements are taken vertically from dangerous positions inside a tornado. Multiple-elevation mobile radar data coupled with photogrammetry techniques and or ground-based wind measurements have successfully derived information on the three-dimensional winds of tornadoes (Wakimoto et al., 2011; Kosiba and Wurman, 2010, 2013; Tanamachi et al., 2013). However, mobile radar-based analyses of winds inside tornadoes using multiple elevations are limited in spatial and temporal resolution, range and provide limited information on the thermodynamics of tornadoes (Markowski et al., 2018)."

Line 67: "[…]of Markowski et al. (2018) and Bartos et al. (2022)".

Corrected

Line 159: :"Federal Aviation Administration (2022) […].

Corrected

Line 167: "[…] (National Association of Rocketry, 2022a, b)."

Corrected

Line 185: "-20oC".

Corrected

Line 196: Time should be in UTC.

Corrected

Please note that the above corrections have been made to the manuscript and will be uploaded later as per the AMT review process for an external preprint.

---

## Referee Report (RR1)

Review: Design and Rocket Deployment of a Trackable Pseudo-Lagrangian Drifter based Meteorological Probe into the Lawrence/Linwood EF4 Tornado and Mesocyclone on 28 May 2019

Authors: Reed Timmer, Mark Simpson, Sean Schofer, and Curtis Brooks

Review Round 2

Overall comments:

In general I thank the authors for their extensive work in addressing the various comments made by all of the reviewers. The effort and dedication to working with feedback is often a considerable effort. In the first review I provided numerous comments, some major and some minor, that I felt would help the manuscript in its presentation of the data collected. Nearly all of these comments have been addressed satisfactorily. I have a few remaining comments to make that I feel should be addressed in the manuscript before publication.

Major comments:

1). In the first review I made several comments on the design of the probe, asking for more detail and documentation of testing. I agree with the authors that adding all of this detail can add significant length to the manuscript. While I personally would rather see these details included directly in the manuscript to demonstrate a rigorous procedure in validating the data, I am ok with the author's decision to provide this information in an external resource that is available to the reader.

That being said, I still have some issue with the presentation of the thermal data from the probe. While the author's answers to my questions/points are reasonable, the fact still remains that the thermal data is significantly lagged compared to the ambient environment. While mathematically a lag correction can be applied to the data, I strongly question the applicability of such a correction in this context given the magnitude of the correction. While radiosondes are generally corrected using a thermal lag treatment as the author's point out, it is on the order of a few degrees rather than ~ 75 degrees as is the case here. The sheer fact that the ambient flow rate inside the probe over the sensor is around 0.17 ms$^{-1}$ is concerning given that it essentially disconnects the probe from the ambient environment. With these factors in mind I have strong reservations about utilizing the thermodynamic data for any scientific purpose.

My preference and suggestion would be to remove presentation of the thermodynamic data from this manuscript. The authors mention changes to the probe design that will be utilized going forward with the sensors mounted on the exterior of the probe, thus making the data significantly more connected to the ambient environment in the future which is good. However, if the authors feel that the data shown here is still worthy of publication (which they do), then I suggest that a statement expressing the limited utility of the data be included to indicate that there are some strong assumptions being made regarding the thermo data and that it is largely being shown for completeness and to illustrate the intent behind the probes mission. Something along the lines of:

"While the thermodynamic data presented here is interesting and unique, it is important to note that significant corrections were required in order to align the data with reasonably expected values. In this correction are several assumptions that may or may not be valid in these conditions. Furthermore, the correction applied to the data is considerable and is largely beyond the level to which corrections are generally applied. With this in mind, careful consideration of the thermodynamic data is warranted and the data presented here is largely done so for illustrative purposes to showcase the potential of the

probe. Future designs of the probe will ideally minimize these potential error sources and lead to more representative data without the need for excessive correction."

I think this could easily fit in the conclusion section. I believe that this or a similar statement which gives the reader caution of utilizing the presented thermo data too much, is a good middle ground in regards to the author's hard work in collecting and publishing their findings, and my reservations regarding the magnitude of the correction. If such a statement is included, I believe the manuscript is ready for publication and I commend the authors in their efforts to work through the review process, as grueling as it can be.

Minor comments:
Line 299-301: I would add a statement at this point explaining that the additional experimental data can be found there which further describes the performance of the probe. The current text doesn't explicitly state that and could be missed by a reader. Something like this following the initial sentence:
"Data from these controlled experiments, including additional documentation as to the process used in testing can be found in an additional resource as cited by Simpson and Timmer (2019)."

---

## Author Response (AR2)

**Reply to Anonymous Referee #3**

The authors appreciate the time and effort that Anonymous Referee #3 has dedicated to providing extensive and valuable feedback on our manuscript to improve the work. We are grateful for your continued insightful comments on our paper.

Major comments:

1). In the first review I made several comments on the design of the probe, asking for more detail and documentation of testing. I agree with the authors that adding all of this detail can add significant length to the manuscript. While I personally would rather see these details included directly in the manuscript to demonstrate a rigorous procedure in validating the data, I am ok with the author's decision to provide this information in an external resource that is available to the reader.

That being said, I still have some issue with the presentation of the thermal data from the probe. While the author's answers to my questions/points are reasonable, the fact still remains that the thermal data is significantly lagged compared to the ambient environment. While mathematically a lag correction can be applied to the data, I strongly question the applicability of such a correction in this context given the magnitude of the correction. While radiosondes are generally corrected using a thermal lag treatment as the author's point out, it is on the order of a few degrees rather than ~ 75 degrees as is the case here. The sheer fact that the ambient flow rate inside the probe over the sensor is around 0.17 ms-1 is concerning given that it essentially disconnects the probe from the ambient environment. With these factors in mind I have strong reservations about utilizing the thermodynamic data for any scientific purpose.

My preference and suggestion would be to remove presentation of the thermodynamic data from this manuscript. The authors mention changes to the probe design that will be utilized going forward with the sensors mounted on the exterior of the probe, thus making the data significantly more connected to the ambient environment in the future which is good. However, if the authors feel that the data shown here is still worthy of publication (which they do), then I suggest that a statement expressing the limited utility of the data be included to indicate that there are some strong assumptions being made regarding the thermo data and that it is largely being shown for completeness and to illustrate the intent behind the probes mission. Something along the lines of:

"While the thermodynamic data presented here is interesting and unique, it is important to note that significant corrections were required in order to align the data with reasonably expected values. In this correction are several assumptions that may or may not be valid in these conditions. Furthermore, the correction applied to the data is considerable and is largely beyond the level to which corrections are generally applied. With this in mind, careful consideration of the thermodynamic data is warranted and the data presented here is largely done so for illustrative purposes to showcase the potential of the probe. Future designs of the probe will ideally minimize these potential

error sources and lead to more representative data without the need for excessive correction."

I think this could easily fit in the conclusion section. I believe that this or a similar statement which gives the reader caution of utilizing the presented thermo data too much, is a good middle ground in regards to the author's hard work in collecting and publishing their findings, and my reservations regarding the magnitude of the correction. If such a statement is included, I believe the manuscript is ready for publication and I commend the authors in their efforts to work through the review process, as grueling as it can be.

We appreciate the reviewer's suggested text, and have added it with some minor amendments to the end of the conclusion, including specifying the temperature data, as pressure was not impacted and relative humidity data was excluded anyway:

"While the thermodynamic data presented here is interesting and unique, it is important to note that significant corrections were required to align the temperature data to reasonably expected values, which was due to the lack of airflow past the sensor in pseudo-Lagrangian flight. In this correction, several assumptions were made that may not be valid in these conditions. Furthermore, the correction applied to the data is considerable and is largely beyond the level to which corrections are generally applied. Careful consideration of the temperature data is warranted and the data presented here is largely for illustrative purposes to showcase the potential of the probe. A future design of the probe minimizes this potential error source without the need for this level of data treatment."

Minor comments:

Line 299-301: I would add a statement at this point explaining that the additional experimental data can be found there which further describes the performance of the probe. The current text doesn't explicitly state that and could be missed by a reader. Something like this following the initial sentence:
"Data from these controlled experiments, including additional documentation as to the process used in testing can be found in an additional resource as cited by Simpson and Timmer (2019)."

This has been changed to:

"Controlled experiments (Fig. 13) were conducted to compute the thermal time constant ($\tau$) of the probe within the enclosure, the probe outside the enclosure, and the self-heating effect. A fan, handheld anemometer, deep freezer, and ice-bath calibrated thermocouple were used for this purpose. Data from these controlled experiments, including additional documentation as to the processes used in testing can be found in the following citation: (Simpson and Timmer, 2019)."